# Fast Algorithms for Packing Proportional Fairness and its Dual

**Francisco Criado**[*]
TU Berlin
Berlin, Germany
criado@math.tu-berlin.de

**David Martínez-Rubio**[*]
Zuse Institute Berlin and TU Berlin
Berlin, Germany
martinez-rubio@zib.de

**Sebastian Pokutta**
Zuse Institute Berlin and TU Berlin
Berlin, Germany
pokutta@zib.de

## Abstract

The proportional fair resource allocation problem is a major problem studied in flow control of networks, operations research, and economic theory, where it has found numerous applications. This problem, defined as the constrained maximization of $\sum_i \log x_i$, is known as the packing proportional fairness problem when the feasible set is defined by positive linear constraints and $x \in \mathbb{R}^n_{\geq 0}$. In this work, we present a distributed accelerated first-order method for this problem which improves upon previous approaches. We also design an algorithm for the optimization of its dual problem. Both algorithms are *width-independent*.

## 1 Introduction

The assignment of bounded resources to several agents under some notions of fairness is a topic studied in networking, operations research, game theory, and economic theory. The allocation obtained by the maximization of the function $\sum_{i=1}^{n} \log(x_i)$ over a convex set $C \subseteq \mathbb{R}^n_{\geq 0}$, known as a *proportional fair allocation*, is an important solution that arises under a natural set of fairness axioms [BFT11; Lan+10]. It corresponds to Nash bargaining solutions [Nas50] and it also has applications to multi-resource allocation in compute clusters [BR15; JH18; Joe+12], rate control in networks [Kel97] and game theory [JV10; JV07]. Other important allocations are linear objectives (no fairness), the max-min allocations [MW00], or $\alpha$-fair allocations [Atk70; MW00; McC+14], which generalize all of the others. Proportional fairness corresponds to $\alpha = 1$. A natural restriction, that many of these applications require, are positive linear constraints. This results in the *packing proportional fairness problem*, also known as the 1-*fair packing problem*. The main focus of this paper is on solving this problem and its dual via first-order methods. Given $A \in \mathcal{M}_{m \times n}(\mathbb{R}_{\geq 0})$, the 1-fair packing problem is

$$\max_{x \in \mathbb{R}^n_{\geq 0}} \left\{ f(x) \overset{\text{def}}{=} \sum_{i=1}^{n} \log x_i : Ax \leq \mathbb{1}_m \right\}. \tag{1FP}$$

We also study the optimization of its Lagrange dual, that can be formulated, cf. Lemma A.1, as

$$\min_{\lambda \in \Delta^m} \left\{ g(\lambda) \overset{\text{def}}{=} -\sum_{i=1}^{n} \log(A^T \lambda)_i - n \log n \right\}, \tag{1FP-Dual}$$

---

[*] Equal contribution.

Most of the notations in this work have a link to their definitions. For example, if you click or tap on any instance of $e_i$, you will jump to the place where it is defined as the $i$-th vector of the canonical base.

36th Conference on Neural Information Processing Systems (NeurIPS 2022).

where $\Delta^m \stackrel{\text{def}}{=} \{\lambda \in \mathbb{R}^m : \sum \lambda_i = 1, \lambda \geq 0\}$ is the $m$-dimensional (probability) simplex. We focus on *width-independent* algorithms that additively $\varepsilon$-approximate the optimum of those problems. For the 1-fair packing problem that means, respectively, that we can find $\bar{x}$ in time that depends at most polylogarithmically on the width $\rho$ of the matrix $A$, and that it satisfies $f^* - f(\bar{x}) \leq \varepsilon$, where $f^*$ is the optimal value. Note that (1FP) has a unique optimizer, by strong concavity. By the same reason, for every two minimizers $\lambda_1, \lambda_2$ of (1FP-Dual), we have $A^T \lambda_1 = A^T \lambda_2$. The width $\rho$ of $A$ is defined as $\max\{A_{ij}\}/\min_{A_{ij} \neq 0}\{A_{ij}\}$, the maximum ratio of the non-zero entries of $A$. Note that in general width-dependent algorithms are not polynomial. Smoothness and Lipschitz constants of the objectives do not scale polylogarithmically with $\rho$ and thus, direct application of classical first-order methods leads to non-polynomial algorithms. As in packing and covering LP, an approximate solution for our primal problem does not necessarily yield one for the dual problem, cf. [AK08], so we need to study them separately. The current form of our techniques does not generalize to $\alpha$-fair problems with $\alpha \neq 1$, but generalizing them to these settings is an interesting future direction of research. We note that previous works treat $\alpha$ in $[0, 1)$, $\{1\}$, or $(1, \infty)$ separately, due to the structure of the problems being different. Most works dealing with $\alpha$-fair functions assume, without loss of generality, that $A$ is given so that the minimum non-zero entry of $A$ is 1 and the maximum entry is $\rho$. However, in this work, we assume without loss of generality that

$$\max_{i \in [m]}\{A_{ij}\} = 1, \text{ for all } j \in [n]. \tag{1}$$

We can do so because, for our problem, we can rescale each primal coordinate multiplicatively, rescaling the columns of $A$ accordingly, which only changes the objectives by an additive constant. Thus, the additive guarantees we will obtain are also satisfied in the non-scaled problem.

Our primal algorithm solves the problem in a distributed model of computation with $n$ agents. Each agent $j \in [n]$ controls variable $x_j$ and only has access to global parameters like $m, n$, or the target accuracy $\varepsilon$, to the $j$-th column of $A$, and in each round it receives the slack $(Ax)_i - 1$ of all the constraints $i$ in which $j$ participates. This is a standard distributed model of computation. We refer to [KY14; AK08] for its motivation and applications.

**Notations** We let $e_i$ be the vector with 1 in coordinate $i$ and 0 elsewhere. We denote by $A_i$ a row of $A$. For $k \in \mathbb{N}$, we use the notation $[k] \stackrel{\text{def}}{=} \{1, 2, \ldots, k\}$. Throughout this work, $\log(\cdot)$ represents the natural logarithm. For $v \in \mathbb{R}^n$, the notation $\exp(v)$ means entrywise exponential. We use $\odot$ for the entrywise product. Given a 1-strongly convex map $\psi$, we denote its Bregman divergence by $D_\psi(x, y) \stackrel{\text{def}}{=} \nabla\psi(x) - \nabla\psi(y) - \langle\nabla\psi(y), x - y\rangle$. We denote by $N$ the number of non-zero entries of the matrix $A$. The notation $\widetilde{O}(\cdot)$ hides logarithmic factors with respect to $m$, $n$, $1/\varepsilon$ and $\rho$. But note that the rates of our algorithms do not depend on $\rho$.

**Related Work** Despite the importance and widespread applicability of fairness objectives, width-independent (and thus polynomial) algorithms for many $\alpha$-fair packing problems were not developed until recently. Width-independent algorithms were first designed for 0-fair packing, i.e., for packing linear programming (LP), that have a longer history [LN93]. For this problem there are currently nearly linear-time width-independent iterative algorithms [AO19] and distributed algorithms [AO15; DO17]. [MSZ16] studied the width-independent optimization of $\alpha$-fair packing problems for any $\alpha \in [0, \infty]$ with a stateless algorithm and [DFO20] gave better rates with a non-stateless algorithm. Both works use the same distributed framework as ours. For the particular case of 1-fair packing, the latter work obtains an unaccelerated algorithm that runs in $\widetilde{O}(n^2/\varepsilon^2)$ distributed iterations. [Bec+14] study the optimization of the dual problem by using Nesterov's accelerated method, and then they reconstruct a primal solution. However, both primal and dual solutions depend on the smoothness constant of the dual problem, which in the worst case is proportional to $\rho^2$, and therefore it is not a polynomial algorithm. In contrast, our algorithms do not depend on $\rho$ at all. Obtaining a priori lower bounds on each of the coordinates of the optimizer is of theoretical and practical interest, since it provides certain amount of resource that can be assigned to each agent before solving the problem. These were studied in [MSZ16] and were improved by [All+18]. In Lemma B.1, we show a lower bound of this kind for our problem when it is normalized as in (1).

**Contribution and Main Results** Our contribution can be summarized as follows; See Table 1 for a comparison with previous works.

*Accelerated algorithm for* 1-*fair packing.* We design a distributed accelerated algorithm for 1-fair packing by generalizing and extending an accelerated technique, designed for packing LP, that

Table 1: Comparison of algorithms for 1-fair packing and its dual. The work of one iteration is linear in $N$, the number of non-zero entries in $A$.

| Paper | Problem | Iterations | Width-dependence? |
|---|---|---|---|
| [Bec+14] | Primal | $O(\rho^2 mn/\varepsilon)$ | Yes |
| [MSZ16] | Primal | $\widetilde{O}(n^5/\varepsilon^5)$ | nearly No (polylog) |
| [DFO20] | Primal | $\widetilde{O}(n^2/\varepsilon^2)$ | nearly No (polylog) |
| **This paper** (Theorem 2.5) | Primal | $\widetilde{O}(n/\varepsilon)$ | No |
| [Bec+14] | Dual | $O(\rho\sqrt{mn/\varepsilon})$ | Yes |
| **This paper** (Theorem 3.4) | Dual | $\widetilde{O}(n^2/\varepsilon)$ | No |

uses truncated gradients of a regularized objective [AO19]. In contrast with this technique, ours yields an algorithm and guarantees that are deterministic. We exploit the structure of our problem to obtain a distributed solution, while for packing LP obtaining a distributed or just parallel algorithm that is accelerated and width-independent is an open question [DO17]. We make use of a different regularization and an analysis that yields additive error guarantees as opposed to multiplicative ones.

*The dual problem.* We consider the dual of the 1-fair packing problem. We reduce the problem to optimizing a proxy function by using the Plotkin-Shmoys-Tardos (PST) framework [PST95; AHK12] with a novel geometric separation oracle. Critical to obtaining fast convergence is showing that the oracle parameters decrease when we obtain better solutions. This fact allows to reduce the dependence on $\varepsilon$, and as a result, our width-independent algorithm enjoys a convergence rate of $\widetilde{O}(n^2/\varepsilon)$ iterations for this problem.

## 2 A distributed accelerated algorithm for 1-Fair Packing

In this section, we present the main steps of our algorithm for the primal problem, which is a deterministic accelerated descent method that optimizes an objective coming from the 1-fair packing problem, and that encodes the constraints in the form of a barrier. Our algorithm approximates the objective additively and allows to compute each iteration in a distributed manner. We note that [DFO20] also made use of this objective for the 1-fair packing problem with different constants, but as opposed to their solution, we allow to compute unfeasible solutions during the course of the algorithm, and we proceed with different techniques that allow to achieve acceleration and thus an algorithm with better convergence rates. We defer the proofs to Appendix B. We reparametrize Problem (1FP) so that the objective function is linear at the expense of making the constraints more complex. That is, we define the function $\hat{f} : \mathbb{R}^n \to \mathbb{R}, x \mapsto f(\exp(x)) = \langle \mathbb{1}_n, x \rangle$. The optimization problem becomes

$$\max_{x \in \mathbb{R}^n} \left\{ \hat{f}(x) \stackrel{\text{def}}{=} \langle \mathbb{1}_n, x \rangle : A \exp(x) \leq \mathbb{1}_m \right\}. \tag{2}$$

Then, we regularize the negative of the reparametrized objective by adding a fast-growing barrier:

$$f_r(x) \stackrel{\text{def}}{=} -\langle \mathbb{1}_n, x \rangle + \frac{\beta}{1+\beta} \sum_{i=1}^m (A \exp(x))_i^{\frac{1+\beta}{\beta}}, \nabla_j f_r(x) = -1 + \sum_{i=1}^m (A \exp(x))_i^{\frac{1}{\beta}} a_{ij} \exp(x_j),$$

where $\beta \stackrel{\text{def}}{=} \frac{\varepsilon}{6n \log(2mn^2/\varepsilon)}$. In this way, we can work with an unconstrained minimization problem. The resulting function is not globally smooth but when the absolute value of a coordinate of the gradient is large, it is positive, and in that case we are able to take a small gradient descent step and decrease the function considerably. The intuition is that if the gradient is large, then the function value along the segment of the gradient step, as a function of the step, can decrease fast. But it cannot increase fast since there are no large negative gradient coordinates. We depict $f_r$ in Figure 1 in Appendix B. The barrier also allows to maintain almost feasibility, as we show in Proposition 2.1 below. It is chosen to grow fast enough so that a point satisfying $(A \exp(x))_i > 1 + \varepsilon/n$, for some $i \in [n]$, will have an optimality gap that is greater than the required accuracy. On the other hand, the regularizer is very small in the feasible region that is not too close to the boundary.

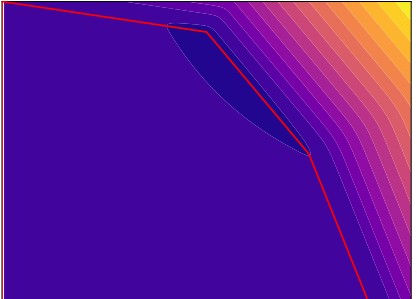 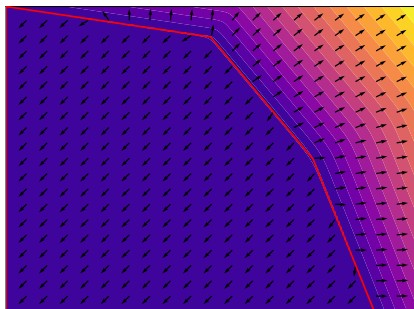

Figure 1: Regularized objective $f_r$ (left) and its gradient (right), for a sample matrix $A \in \mathcal{M}_{3\times 2}(\mathbb{R}_{\geq 0})$. For visualization purposes we show $\log(f_r(x))$ and $\log(\|\nabla f_r(x)\|)$, represented by color, and we indicate the direction of the gradient with normalized arrows. Also, note that we show the results in the original space (i.e., before reparametrizing, so the constraints appear to be linear) but the gradient was computed as originally defined (i.e., after reparametrizing).

Let $\hat{x}^*$ be the maximizer of $\hat{f}$, let $x^* \stackrel{\text{def}}{=} \exp(\hat{x}^*)$ be the solution to Problem (1FP), and let $x_r^*$ be the minimizer of $f_r$. We have $\hat{x}^* \in [-\log(n), 0]^n$ by Lemma B.1. Let $\omega \stackrel{\text{def}}{=} \log(mn/(1-\varepsilon/n))$ and define the box $B \stackrel{\text{def}}{=} [-\omega, 0]^n$. We restrict ourselves to this domain and formulate our final problem, that we will minimize with an accelerated method:

$$\min_{x \in B} f_r(x). \tag{1FP-primalReg}$$

Note $f_r(x) \geq 0$ if $x \in B$. We add the redundant and simple box constraints $B$ in order to later guarantee a bound on the regret of the mirror descent method that runs within the algorithm. We show that it suffices to obtain an $\varepsilon$-minimizer of Problem (1FP-primalReg) in order to obtain an $O(\varepsilon)$-minimizer for the original Problem (1FP).

**Proposition 2.1.** [↓] *Let $\varepsilon \in (0, n/2)$. Let $x_r^*$ be the minimizer of (1FP-primalReg) and let $x_r^\varepsilon \in B$ be an $\varepsilon$-minimizer of this problem. Then the point $\bar{u} \stackrel{\text{def}}{=} \exp(x_r^\varepsilon)/(1+\varepsilon/n)$ satisfies $f(x^*) - f(\bar{u}) \leq 5\varepsilon$ and $A\bar{u} \leq \mathbb{1}_m$, where $x^*$ is the maximizer of $f$.*

The intuition about this proposition is that $x_r^\varepsilon$ is also a point with low $\hat{f}$ value. By the aforementioned barrier guarantees, it is almost feasible, i.e., $A\exp(x_r^\varepsilon) \leq 1 + \varepsilon/n$, and dividing the corresponding $\exp(x_r^\varepsilon)$ by $1 + \varepsilon/n$, and thus making it feasible, can only increase the objective $f$ by $\varepsilon$.

In the sequel, we will present the different parts of Algorithm 1 and their analyses. In particular, the notation and definitions used are compatible with the choices in the algorithm and most of the parameter choices naturally occur throughout the arguments. Our optimization algorithm starts at the points $x^{(0)} = y^{(0)} = z^{(0)} = -\log(mn/(1-\varepsilon/n))\mathbb{1}_n$ and updates each of these variables $x^{(k)}, y^{(k)}$ and $z^{(k)}$ once in each iteration. They remain in $B$, cf. Lemma B.2. The role of the three variables is the following: $z^{(k)}$ will be a mirror point and $y^{(k)}$ will be a gradient descent point, in the sense that in order to compute them we apply mirror descent and gradient descent. Then, the point $x^{(k)}$ will be a convex combination of both, that will balance the regret of $z^{(k)}$ with the primal progress of $y^{(k)}$, effectively coupling these two algorithms.

It is important to note that we do not use the gradient $\nabla f_r(x)$ for our mirror descent loss. Instead, we use a truncation of the gradient. More precisely, the loss we perform the mirror descent step on is the truncated gradient $\overline{\nabla f_r}(x^{(k)}) \in \mathbb{R}^n$ defined as

$$\overline{\nabla_i f_r}(x^{(k)}) \stackrel{\text{def}}{=} \min\{1, \nabla_i f_r(x^{(k)})\} \text{ for all } i \in [n]. \tag{3}$$

Note that $\overline{\nabla f_r}(x^{(k)}) \in [-1, 1]^n$ because $\nabla f_r(x) \in [-1, \infty]^n$ for any $x \in \mathbb{R}^n$, as the regularizer has positive gradient; see also definition of $f_r(x)$ and its gradient. The truncation allows mirror descent

to control one part of the regret, which will not depend on the global Lipschitz constant. Gradient descent will compensate for both such regret and the part that is not controlled by mirror descent.

Let $\Pi_{\mathcal{X}}(\cdot)$ be the $\|\cdot\|_2$-projection map of a point onto a convex set $\mathcal{X}$. The mirror descent update can be written in closed form as any of the two following equivalent ways

$$
\begin{aligned}
z^{(k)} &\leftarrow \Pi_B(z^{(k-1)} - \omega\eta_k\overline{\nabla f_r}(x^{(k)})), \\
z_i^{(k)} &\leftarrow \Pi_{[-\omega,0]}(z_i^{(k-1)} - \omega\eta_k\overline{\nabla_i f_r}(x^{(k)})), \text{ for all } i \in [n].
\end{aligned}
\tag{4}
$$

That is, projecting back to the box, in case of the $\|\cdot\|_2$, consists of simply clipping each coordinate. We bound the regret coming from this mirror descent step by modifying the classical analysis of mirror descent, cf. Lemma B.3.

**Lemma 2.2 (Mirror Descent Guarantee).** [↓] *Let $u \in B$ and choose $L$ as in Algorithm 1. We have:*

$$
\langle\eta_k\overline{\nabla f_r}(x^{(k)}), z^{(k-1)} - u\rangle \leq \eta_k^2 L\langle\overline{\nabla f_r}(x^{(k)}), x^{(k)} - y^{(k)}\rangle + \frac{1}{2\omega}\|z^{(k-1)} - u\|_2^2 - \frac{1}{2\omega}\|z^{(k)} - u\|_2^2.
$$

Next, we will analyze the role of the gradient descent step. We show in the following lemma a lower bound on the progress of our descent step. Note that this progress could not be greater than $\langle\nabla f_r(x^{(k)}), x^{(k)} - y^{(k)}\rangle$, by convexity, so this is a strong descent condition.

**Lemma 2.3 (Descent Lemma).** [↓] *Given $x^{(k)}$ and $y^{(k)}$ as defined in Algorithm 1, the following holds:*

$$
f_r(x^{(k)}) - f_r(y^{(k)}) \geq \frac{1}{2}\langle\nabla f_r(x^{(k)}), x^{(k)} - y^{(k)}\rangle \geq 0.
$$

---

**Algorithm 1** Accelerated descent method for 1-Fair Packing

---

**Input:** Matrix $A \in \mathcal{M}_{m\times n}(\mathbb{R}_{\geq 0})$ normalized as in (1). Accuracy $\varepsilon \in (0, n/2]$.

1: $\beta \leftarrow \frac{\varepsilon}{6n\log(2mn^2/\varepsilon)}$; $\omega \leftarrow \log(\frac{mn}{1-\varepsilon/n})$; $L = \max\left\{\frac{4\omega(1+\beta)}{\beta}, \frac{16n\log(2mn)}{3\varepsilon} + \frac{1}{3}\right\} = \widetilde{O}(n/\varepsilon)$

2: $\eta_0 \leftarrow \frac{1}{3L}$; $C_k = 3\eta_k L$; $\tau \leftarrow \tau_k = \eta_k/C_k = \frac{1}{3L}$.

3: $T \leftarrow \lceil\log(\frac{4n\log(2mn)}{\varepsilon})/\log(\frac{1}{1-\tau})\rceil \leq \lceil 3L\log(\frac{4n\log(2mn)}{\varepsilon})\rceil = \widetilde{O}(n/\varepsilon)$

4: $x^{(0)} \leftarrow y^{(0)} \leftarrow z^{(0)} \leftarrow -\omega\mathbb{1}_n$

---

5: **for** $k = 1$ **to** $T$ **do**

6:     $\eta_k \leftarrow C_k - C_{k-1} = \frac{1}{1-\tau}\eta_{k-1}$

7:     $x^{(k)} \leftarrow \tau z^{(k-1)} + (1-\tau)y^{(k-1)}$

8:     $z^{(k)} \leftarrow \arg\min_{z\in B}\left\{\frac{1}{2\omega}\|z - z^{(k-1)}\|_2^2 + \langle\eta_k\overline{\nabla f_r}(x^{(k)}), z\rangle\right\}$     ⋄ Mirror descent step

9:     $y^{(k)} \leftarrow x^{(k)} + \frac{1}{\eta_k L}(z^{(k)} - z^{(k-1)})$     ⋄ Gradient descent step

10: **end for**

11: **return** $\bar{x} \stackrel{\text{def}}{=} \exp(y^{(T)})/(1 + \varepsilon/n)$

**Output:** $f(\bar{x}) - f(x^*) \leq \varepsilon$ and $\bar{x}$ is feasible, i.e., $A\bar{x} \leq 1$. The total number of iterations is $\widetilde{O}(n/\varepsilon)$ to obtain an $O(\varepsilon)$-approximate solution.

---

### 2.1 Coupling Mirror Descent and Gradient Descent

We first prove a lemma that shows we can compensate for the regret coming from mirror descent as well as for the rest of the regret. Note the total weighted instantaneous regret $\langle\eta_k\nabla f(x^{(k)}), z^{(k-1)} - u\rangle$ is bounded by the left hand side of (5) up to a difference of potential functions, by Lemma 2.2. This is a critical part of the analysis: using the truncated gradient for mirror descent makes its corresponding regret not to depend on the smoothness constant, but there is a remaining regret that, crucially, can be compensated by our strong descent condition.

**Lemma 2.4.** [↓] *Let $C_k \stackrel{\text{def}}{=} 3\eta_k L$, and let $\nu^{(k)} \stackrel{\text{def}}{=} \nabla f_r(x^{(k)}) - \overline{\nabla f_r}(x^{(k)}) \in [0, \infty)^n$. For all $u \in B$, we have*

$$
\langle\eta_k\nu^{(k)}, z^{(k-1)} - u\rangle + \eta_k^2 L\langle\overline{\nabla f_r}(x^{(k)}), x^{(k)} - y^{(k)}\rangle \leq C_k(f_r(x^{(k)}) - f_r(y^{(k)})).
\tag{5}
$$

With these tools at hand, we can now use a linear coupling argument to establish an accelerated convergence rate. Note that the algorithm takes the simple form of iterating a mirror descent step, gradient descent step and a coupling, after a careful choice of parameters. All of which depend on known quantities.

**Theorem 2.5.** [↓] *Let $\varepsilon \le n/2$ and let $x^*$ be the solution to* (1FP) *and let $x_r^*$ be the minimizer of* (1FP-primalReg). *Algorithm 1 computes a point $y^{(T)} \in B$ such that $f_r(y^{(T)}) - f_r(x_r^*) \le \varepsilon$ in a number of iterations $T = \widetilde{O}(n/\varepsilon)$. Besides, $\bar{x} \stackrel{\text{def}}{=} \exp(y^{(T)})/(1 + \varepsilon/n)$ is a feasible point of* (1FP), *i.e., $A\bar{x} \le \mathbb{1}_m$, such that $f(x^*) - f(\bar{x}) \le 5\varepsilon = O(\varepsilon)$.*

## 3 Optimizing the dual problem

In this section, we reduce the dual problem (1FP-Dual) to the feasibility problem of a packing LP, which we call the *proxy* problem. Then, we show how we can use the PST framework [PST95] to approximately solving the proxy. Our algorithm relies on a carefully built geometric oracle whose *width parameters* can be guaranteed to decrease with the optimality gap. We perform a sequence of restarts, and after each of them we can guarantee lower oracle width. This allows for reducing the overall complexity.

### 3.1 The dual 1-fair packing problem as a feasibility packing LP

Let $\mathcal{P} \stackrel{\text{def}}{=} \{x \in \mathbb{R}_{\ge 0}^n : Ax \le \mathbb{1}_m\}$ be the feasible region of the primal problem (1FP). In this section, we identify (non-negative) covering constraints of the form $\langle h, x \rangle \le 1$ with vector $h \in \mathbb{R}_{\ge 0}^n$. Recall we assume without loss of generality that $A$ satisfies (1), that is, the maximum entry of each column is 1. This implies that $e_i \in \mathcal{P}$ for all $i \in [n]$, and $\mathcal{P} \subseteq [0,1]^n$. For this reason, we also assume in this section and without loss of generality that $A$ contains the constraints $\{e_i\}_{i=1}^n$, i.e., $x_i \le 1$, for $i \in [n]$. For convenience, we assume they are the first $n$ rows of $A$.

Let $\lambda^* \in \Delta^m$ be an optimal solution of Problem (1FP-Dual) and let $h^{\lambda^*} \stackrel{\text{def}}{=} A^T\lambda^* \in \mathbb{R}_{\ge 0}^n$. By strong duality of the Lagrange dual, we can reconstruct the optimal primal solution as $x^* = (1/(nh_1^{\lambda^*}), \ldots, 1/(nh_n^{\lambda^*}))$. This motivates the definition of the *centroid map*:

$$c(h) \stackrel{\text{def}}{=} \left( \frac{1}{nh_1}, \ldots, \frac{1}{nh_n} \right); \quad c^{-1}(x) \stackrel{\text{def}}{=} \left( \frac{1}{nx_1}, \ldots, \frac{1}{nx_n} \right),$$

where $h, x \in \mathbb{R}_{\ge 0}^n$ are constraints and points, respectively. Despite the two maps above being the same function we distinguish between $c$ and $c^{-1}$ to unambiguously refer to constraints or points. The name of the centroid map is motivated by the fact that, for any constraint $h \in \mathbb{R}_{\ge 0}^n$, the point $c(h)$ is the geometric centroid of the simplex $\{x \in \mathbb{R}_{\ge 0}^n : \langle h, x \rangle = 1\}$. Given a $\lambda \in \Delta^m$, we define its associated constraint $h^\lambda \stackrel{\text{def}}{=} A^T\lambda$. In addition, we define the centroid $p^\lambda \stackrel{\text{def}}{=} c(A^T\lambda) = c(h^\lambda)$. Note that we can efficiently compute $h^\lambda, p^\lambda$ from $\lambda$, and $h^\lambda$ from $p^\lambda$ and viceversa. However, we cannot obtain the coefficients $\lambda$ efficiently from $h^\lambda$ or $p^\lambda$, as this amounts to solving a linear program.

An important property is that $h^{\lambda^*}$ is the unique constraint of the form $h^\lambda$ such that $c(h^\lambda) \in \mathcal{P}$, for some $\lambda \in \Delta^m$. It is the optimizer because if $h^{\lambda^*}$ has $c(h^{\lambda^*}) \in \mathcal{P}$, then $(p^{\lambda^*}, \lambda^*)$ satisfies the optimality conditions. It is unique because of strong convexity of $-\log(\cdot)$; any other dual optimizer $\bar{\lambda}^*$ will have $A^T\lambda^* = A^T\bar{\lambda}^*$. The following lemma shows an approximate version of this property.

**Lemma 3.1.** [↓] *Let $h^\lambda$ for $\lambda \in \Delta^m$, such that $Ap^\lambda \le (1+\varepsilon)\mathbb{1}_m$. Let $\lambda^*$ be the minimizer of Problem* (1FP-Dual)*, of objective function $g$. Then, $g(\lambda) - g(\lambda^*) \le n\log(1+\varepsilon) \le n\varepsilon$.*

This lemma motivates the minimization of $\max_{i \in [m]}\langle A_i, c(A^T\lambda)\rangle$ for $\lambda \in \Delta^m$ as a proxy for solving Problem (1FP-Dual). Furthermore, if we optimize over the set $\{p^\lambda = c(A^T\lambda) : \lambda \in \Delta^m\}$, we end up with a feasibility problem in a packing LP. However, this set is not convex in general, which is a requirement of the PST framework we intend to use. Fortunately, we can optimize over a larger and convex set while preserving the minimizer. Indeed, define the sets of constraints $\mathcal{D} \stackrel{\text{def}}{=} \{A^T\lambda : \lambda \in \Delta^m\} = \text{conv}(\{A_1, \ldots, A_m\})$ and $\mathcal{D}^+ \stackrel{\text{def}}{=} \{A^T\lambda - \mu \ge \mathbb{0} : \lambda \in \Delta^m, \mu \in \mathbb{R}_{\ge 0}^n\} = \{h \in \mathbb{R}_{\ge 0}^n : h \le h^\lambda, \text{ for } h^\lambda \in \mathcal{D}\}$. For a constraint $h \in \mathbb{R}_{\ge 0}$, we have by polyhedral duality that $h \in \mathcal{D}^+$ if and only if $\langle h, x \rangle \le 1$ for all $x \in \mathcal{P}$. In other words, $\mathcal{D}^+$ is exactly the set

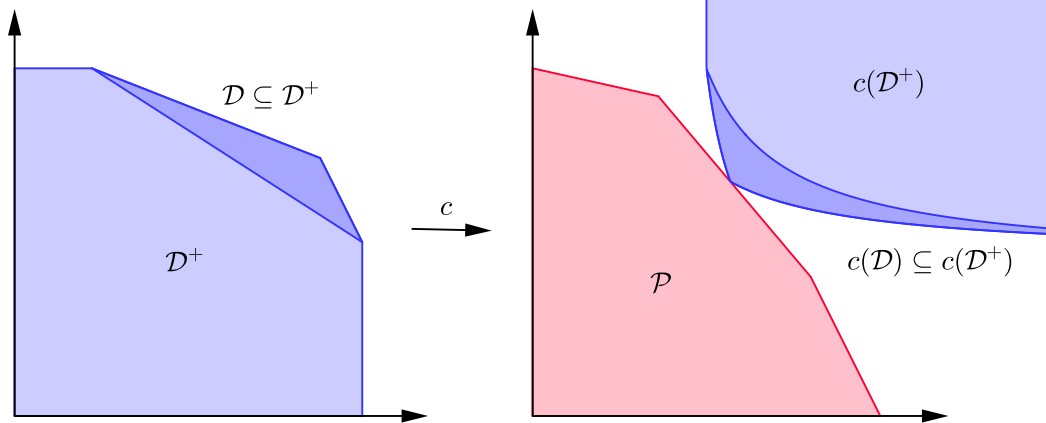

Figure 2: Left, the dual polytope $\mathcal{D}^+$ containing $\mathcal{D}$. The centroid $c(\cdot)$ maps dual points (i.e., constraints) to primal points. Right, $\mathcal{P}$ and the image of $\mathcal{D}$ and $\mathcal{D}^+$ via $c$. Note $c(\mathcal{D}^+) \cap \mathcal{P}$ is exactly one point, which is also contained in $c(\mathcal{D})$. And note $c(\mathcal{D})$ can be non-convex, but $c(\mathcal{D}^+)$ is convex.

of positive constraints $\langle h, x \rangle \leq 1$ with $h \in \mathbb{R}^n_{\geq 0}$ that are feasible for $\mathcal{P}$. We can think about the optimization of Problem (1FP-Dual) as

$$\min_{h^\lambda \in \mathcal{D}} \{-\sum_{i=1}^n \log(h_i^\lambda) - n \log(n)\} \overset{\text{①}}{=} \min_{h \in \mathcal{D}^+} \{-\sum_{i=1}^n \log(h_i) - n \log(n)\},$$

where ① comes from the following observation: since $h \in \mathcal{D}^+$ if and only if there is $h^\lambda \in \mathcal{D}$ with $h \leq h^\lambda$, and the expression $-\sum_{i=1}^n \log(h_i) - n \log(n)$ is strictly decreasing in every $h_i$ then no $h \in \mathcal{D}^+ \setminus \mathcal{D}$ could ever minimize the right hand side. Consequently, the minimizer of both problems is the same. This results in the following proxy, motivated by Lemma 3.1:

$$\min_{p \in c(\mathcal{D}^+)} \left\{ \hat{g}(p) \overset{\text{def}}{=} \max_{i \in [m]} \langle A_i, p \rangle \right\}. \tag{1FPD-Proxy}$$

By Lemma 3.1, the optimizer of this problem must be $p^{\lambda^*}$. Note $\hat{g}(p^{\lambda^*}) = 1$ so we want a $p$ such that $\hat{g}(p) \leq 1 + \varepsilon/n$. We solve this as an approximate feasibility problem in a packing LP by using the PST framework over $c(\mathcal{D}^+)$, which is convex, according to the following lemma.

**Lemma 3.2.** [↓] *Let $p^{(1)}, \ldots, p^{(k)}$ be points in $c(\mathcal{D})$, and let $\zeta \in \Delta^k$ be coefficients. Then, we have* $c(\zeta_1 c^{-1}(p^{(1)}) + \cdots + \zeta_k c^{-1}(p^{(k)})) \leq \zeta_1 p^{(1)} + \cdots + \zeta_k p^{(k)}$. *Consequently, $c(\mathcal{D}^+)$ is convex.*

### 3.2 The PST algorithm

Plotkin, Shmoys, and Tardos designed a framework (PST) for solving LP [PST95]. This result was improved by [AHK12] for the case of packing and covering LP. We can use this framework to solve Problem (1FPD-Proxy) if we can provide a good feasibility oracle, as explained in the sequel. The PST framework focuses on checking the feasibility of $Ap \leq \mathbb{1}_m$, with $p$ in a convex set $\mathcal{X}$. Then, it obtains either a certificate of infeasibility of the problem or computes a point $p \in \mathcal{X}$ such that $Ap \leq (1+\varepsilon)\mathbb{1}_m$, for some given $\varepsilon > 0$. The PST framework works by calling an oracle which solves the simpler feasibility problem of finding

$$p \in \mathcal{X} \text{ such that } \langle A^T \Lambda, p \rangle \leq 1, \tag{6}$$

for some distribution $\Lambda \in \Delta^m$. That is, given a single constraint $h^\Lambda$, written as a convex combination of the constraints defined by $A$, the query asks for a point $p \in \mathcal{X}$ that satisfies the constraint. In our case, we apply the framework to $\mathcal{X} = c(\mathcal{D}^+)$ to find a point $p \in c(\mathcal{D}^+)$ with $Ap \leq (1+\varepsilon)\mathbb{1}_m$. We can apply PST because $c(\mathcal{D}^+)$ is convex. Our oracle subproblems are always solvable because $p^{\lambda^*} \in \mathcal{P}$ satisfies them all. We use the following formulation of the PST and multiplicative weights (MW) algorithms for packing LP, which is a slight variation of [AHK12]. The MW algorithm and its guarantees are presented in Lemma C.1.

---

**Algorithm 2** Optimization of the dual of 1-fair packing with oracle $\mathfrak{O}$

---

**Input:** Matrix $A \in \mathcal{M}_{m \times n}(\mathbb{R}_{\geq 0})$ normalized as in (1). Accuracy $\varepsilon \in (0, (n-1)n]$.

1: $\bar{\lambda}^{(0)} \leftarrow \text{concat}(\mathbb{1}_n/n, \mathbb{0}_{m-n}) \in \Delta^m; \quad \varepsilon_{-1} = n - 1 \qquad \diamond p^{\bar{\lambda}^{(t)}}$ is an $\varepsilon_{t-1}$-minimizer of $\hat{g}$

2: **for** $t = 0$ **to** $T \stackrel{\text{def}}{=} \max\{0, \lceil \log_2(2/(\varepsilon/n)) \rceil\}$ **do**

3: $\quad I_t \leftarrow \{i \in [m] : \langle A_i, p^{\bar{\lambda}^{(t)}} \rangle \geq \frac{1+\varepsilon_{t-1}}{1+2\varepsilon_{t-1}n+\sqrt{2\varepsilon_{t-1}n}}\}$ $\qquad \diamond$ Remove redundant constraints

4: $\quad$ **if** $t$ **is** 0 **then** $\varepsilon_t \leftarrow \max\{2, \varepsilon/n\}$ **else** $\varepsilon_t \leftarrow \varepsilon_{t-1}/2$ $\qquad \diamond$ Next target accuracy

5: $\quad \Lambda^{(1)} \leftarrow \mathbb{1}_{|I_t|}$ $\qquad \diamond$ Restart MW

6: $\quad$ **for** $k = 1$ **to** $K_t \stackrel{\text{def}}{=} 32\tau_{\varepsilon_{t-1}}\sigma_{\varepsilon_{t-1}} \log(|I_t|)/\varepsilon_t^2 = \widetilde{O}(n/\varepsilon_t)$ **do**

7: $\quad\quad \lambda^{(k)}, p^{\lambda^{(k)}} \leftarrow \mathfrak{O}(\text{query } \Lambda^{(k)}/\|\Lambda^{(k)}\|_1, \text{ current solution } \bar{\lambda}^{(t)}, \text{ index set } I_t) \in (\Delta^m, c(\mathcal{D}))$

8: $\quad\quad \ell^{(k)} \leftarrow \mathbb{1}_{|I_t|} - A_{I_t} p^{\lambda^{(k)}}$

9: $\quad\quad \Lambda^{(k+1)} \leftarrow \Lambda^{(k)} \odot (\mathbb{1}_{|I_t|} - (\varepsilon_t/(4\tau_{\varepsilon_{t-1}}\sigma_{\varepsilon_{t-1}})) \cdot \ell^{(k)})$ $\qquad \diamond$ MW step

10: $\quad$ **end for**

11: $\quad \bar{\lambda}^{(t+1)} \leftarrow \frac{1}{K_t} \sum_{k=1}^{K_t} \lambda^{(k)}$

12: **end for**

13: **return** $\bar{\lambda} \stackrel{\text{def}}{=} \bar{\lambda}^{(T+1)}$

**Output:** $Ap^{\bar{\lambda}} \leq (1 + \varepsilon/n)\mathbb{1}_m$, that is, $\hat{g}(p^{\bar{\lambda}}) \leq 1 + \varepsilon/n$. Hence $g(\bar{\lambda}) - g(\lambda^*) \leq \varepsilon$.

---

**Lemma 3.3 (PST guarantee).** [↓] *Let $\sigma, \tau \in \mathbb{R}_{>0}$. For a target accuracy $\varepsilon \in (0, 4\min\{\sigma, \tau\}]$, run the MW algorithm of Lemma C.1 with $\delta = \varepsilon/2$, losses $\ell^{(k)} \stackrel{\text{def}}{=} \mathbb{1}_m - Ap^{(k)}$ assumed to be in $[-\sigma, \tau]^m$, where $p^{(k)}$ is the point the oracle outputs at iteration $k$ when the constraint that is queried is given by $A^T(\Lambda^{(k)}/\|\Lambda^{(k)}\|_1)$, and where $\Lambda^{(k)}$ are the weights computed by the MW algorithm. Then, after $K = \frac{32\sigma\tau \log(m)}{\varepsilon^2}$ iterations, we obtain a solution $\bar{p} \stackrel{\text{def}}{=} \frac{1}{K} \sum_{k=1}^K p^{(k)}$ that satisfies $\hat{g}(\bar{p}) \leq 1 + \varepsilon$.*

In order to give a solution of Problem (1FP-Dual), we are also interested in recovering some $\lambda \in \Delta^m$ such that $\hat{g}(p^\lambda)$ is small. Assume the oracle returns a point $p^{(k)} = p^{\lambda^{(k)}}$ for some $\lambda^{(k)}$ it can also provide. Then, even if $\bar{p} \in \mathcal{D}^+ \setminus \mathcal{D}$, we have by Lemma 3.2 that $\bar{\lambda} \stackrel{\text{def}}{=} \frac{1}{K} \sum_{k=1}^K \lambda^{(k)}$ defines a point $p^{\bar{\lambda}} \in \mathcal{D}$ such that $p^{\bar{\lambda}} \leq \bar{p}$. Hence $\hat{g}(p^{\bar{\lambda}}) \leq \hat{g}(\bar{p}) \leq 1 + \varepsilon$, so $\bar{\lambda}$ can be used as our dual solution.

---

**Algorithm 3** Feasibility oracle $\mathfrak{O}$

---

**Input:** An approximate solution $s \stackrel{\text{def}}{=} A^T \lambda^{(s)}, \lambda^{(s)} \in \Delta^m, \hat{g}(c(s)) \leq 1 + \delta$. Query constraint $q = A^T \lambda^{(q)}, \lambda^{(q)} \in \Delta^m$. Precision parameter $1 < \omega \leq 2$, default value $\omega = 2$. Index set of *non-redundant constraints* $I = \{i \in [m] : \langle A_i, c(s) \rangle \geq \frac{1+\delta}{1+\omega\delta n+\sqrt{\omega\delta n}}\}$.

**Output:** $\lambda^{(o)} \in \Delta^m$ and point $o \stackrel{\text{def}}{=} c(A^T \lambda^{(o)}) \in c(\mathcal{D})$ such that

1. $o$ is covered by $q$, i.e., $\langle q, o \rangle \leq 1$.

2. If $i \in I$ then $\langle A_i, o \rangle \in [1-\tau, 1+\sigma]$ where $\sigma = \min(\sqrt{\omega\delta n}+\omega\delta n, \frac{1+\omega\delta}{1+\delta}\max_{i\in[n]} s_i^{-1}-1)$, $\tau = \min(3\sqrt{\omega\delta n}, 1)$.

3. It satisfies all redundant constraints, i.e., $\langle A_i, o \rangle \leq 1$, if $i \in [m] \setminus I$.

---

1: **if** $\langle s, c(q) \rangle \leq \frac{1+\omega\delta}{1+\delta}$ **then return** $\lambda^{(o)} = \lambda^{(q)}, \quad o = c(q)$

2: **else if** $\langle q, c(s) \rangle \leq 1$ **then return** $\lambda^{(o)} = \lambda^{(s)}, \quad o = c(s)$

3: **else** Find $\mu \in (0, 1)$ s.t. $\langle s, c((1-\mu)s + \mu q) \rangle \in (1, \frac{1+\omega\delta}{1+\delta})$ via the bisection method

4: **end if**

5: **return** $\lambda^{(o)} = (1-\mu)\lambda^{(s)} + \mu\lambda^{(q)}, \quad o = c((1-\mu)s + \mu q)$

---

Thus our task is to construct an oracle with good enough $\sigma$ and $\tau$, which are called the width parameters of the oracle. We also need to make sure that $\varepsilon \leq 4\min\{\sigma, \tau\}$, and that we can output a point $p^\lambda$ in $\mathcal{D}$, and a corresponding $\lambda$. Regardless, this algorithm runs in $\widetilde{O}(\sigma\tau/\varepsilon^2)$ iterations, which is slower than what we aim for, for constant $\sigma$ and $\tau$. Our approach to obtain a fast algorithm is to design an adaptive feasibility oracle. We provide our best solution $\bar{\lambda}^{(t)}$ to the oracle, which

satisfies $\hat{g}(p^{\bar{\lambda}^{(t)}}) \le 1 + \delta$, for some $\delta > 0$. Given such a $\delta$-approximate solution we identify a set of indices $I_t \subseteq [n]$ such that the constraints $\{A_i, i \notin I_t\}$ are redundant, in the sense that the oracle is guaranteed to return points satisfying them even if they are removed from the problem. We remove these constraints since they would yield large values of $\tau$. Hence, we can run our MW algorithm with the remaining $|I_t|$ constraints. Denote $A_{I_t}$ the matrix that has $\{A_i : i \in I_t\}$ as rows, in increasing order of $i$. In this case if $\Lambda \in \Delta^{|I_t|}$, we denote $h^\Lambda = A_{I_t}^T \Lambda$ and $p^\Lambda = c(A_{I_t}^T \Lambda)$ accordingly. Our oracle, when given a query $h^\Lambda$, uses both constraints $h^\Lambda$ and $h^{\bar{\lambda}^{(t)}}$ to return a point $p$ satisfying the oracle condition (6) and such that the loss $\mathbb{1}_{|I_t|} - A_{I_t} p$ is in $[-\sigma_\delta, \tau_\delta]^{|I_t|}$, with

$$\sigma_\delta \overset{\text{def}}{=} \begin{cases} \sqrt{2\delta n} + 2\delta n & \text{if } \delta \le 2 \\ \frac{1+2\delta}{1+\delta} \max_{i \in [n]}\{1/h_i^{\bar{\lambda}^{(t)}}\} - 1 & \text{if } \delta > 2 \end{cases}, \qquad \tau_\delta \overset{\text{def}}{=} \min\{3\sqrt{2\delta n}, 1\}. \qquad (7)$$

In fact, $\sigma_\delta$ can be defined as the minimum of its two expressions above, regardless of the value of $\delta$. But we shall use this definition for our algorithm. In the next section, we present the construction and analysis of such an oracle and the set $I_t$. We observe that, because the parameters depend on $\delta$, we can restart the MW algorithm, and run it in several stages, indexed by $t = 0, \ldots, T$. This allows for incrementally reducing the width parameters and obtaining a better overall complexity, as we prove in the following theorem.

**Theorem 3.4.** [↓] *Let $\varepsilon \in (0, n(n-1)]$. Suppose we have a feasibility oracle $\mathfrak{O}$ satisfying (6) and (7), when we filter constraints according to Algorithm 2. Then, Algorithm 2 computes, in $\widetilde{O}(n^2/\varepsilon)$ iterations, a point $\bar{\lambda}$ which is an $\varepsilon$-minimizer of $g$. Moreover, $p^{\bar{\lambda}}$ is an $(\varepsilon/n)$-minimizer of $\hat{g}$.*

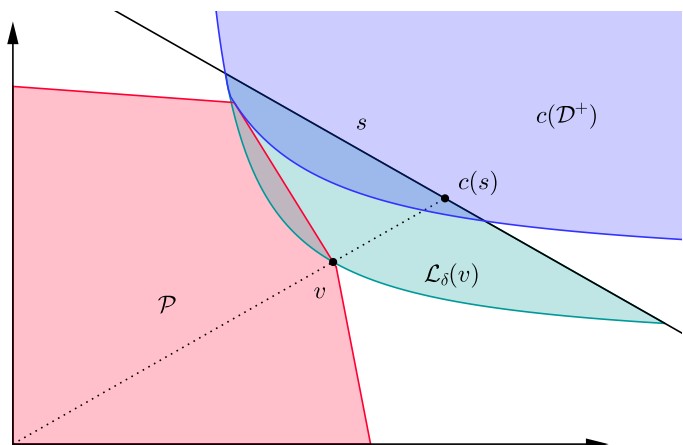

Figure 3: The lens $\mathcal{L}_\delta(v)$ given by a feasible solution $s$, for $\omega = 1$. In the actual algorithm, we have $\omega > 1$, which defines a larger set in which the affine part is translated upwards.

### 3.3 The PST oracle and the redundant constraints

In order to give a fast solution for Problem (1FPD-Proxy), we want to design an adaptive oracle. That is, it should satisfy (6), and it should output better points if it already knows a good approximate solution. Generically, assume that the oracle has access to a feasible constraint $s \overset{\text{def}}{=} A^T \lambda^{(s)} \in \mathcal{D}$, with $\lambda^{(s)} \in \Delta^m$ satisfying $p^{\lambda^{(s)}} = c(s)$ is a point with $\hat{g}(p^{\lambda^{(s)}}) \le 1 + \delta$. In other words, $c(s)$ is a $\delta$-approximate solution to Problem (1FPD-Proxy). Let $q \overset{\text{def}}{=} A^T \lambda^{(q)} \in \mathcal{D}$, with $\lambda^{(q)} \in \Delta^m$ be a query constraint. In Algorithm 2, $\lambda^{(s)}$ and $\lambda^{(q)}$ correspond to $\bar{\lambda}^{(t)}$ and $\Lambda^{(k)}/\|\Lambda^{(k)}\|_1$ respectively, where the latter is interpreted as having coefficients equal to zero for constraints $i \notin I_t$.

The main geometric idea is that by using the solution $s$, we can define a region whose size depends on $\delta$ containing the optimum $p^{\lambda^*}$ of Problem (1FPD-Proxy). We will guarantee the oracle returns points in this region, which in turn means that the oracle returns points close to the optimum. We call this geometric object the *lens of a point*:

$$\mathcal{L}_{\omega\delta}(v) \overset{\text{def}}{=} \{x \in \mathbb{R}_{\ge 0}^n : \langle c^{-1}(x), v \rangle \le 1, \langle c^{-1}(v), x \rangle \le 1 + \omega\delta\},$$

where $\omega \in (1, 2]$ is a parameter; we chose $\omega = 2$ in the algorithm. We depict the lens in Figure 3 in Appendix C. We apply this definition to the point $v \overset{\text{def}}{=} c(s)/(1 + \delta)$, which is in $\mathcal{P}$ because $\hat{g}(s) \leq 1 + \delta$ implies that $c(s)/(1 + \delta) \in \mathcal{P}$.

We show in Lemma C.2 that, by using the bisection method, we can efficiently find a convex combination $(1 - \mu)s + \mu q$ that will result in a constraint, whose centroid is the point $o$ we output. It satisfies the first oracle condition, and is in the lens. Also, we can recover $\lambda^{(o)}$ as $(1 - \mu)\lambda^{(s)} + \mu\lambda^{(q)}$.

If the oracle can output a point $o$ in the lens, we will have low width parameters for the constraints $A_i$ that do not cover the lens, as we show in Proposition C.4. That is, for one such constraint $A_i$, the corresponding loss is $1 - \langle A_i, o \rangle \in [-\sigma, \tau]$. The other constraints could be problematic in terms of width parameters because $\langle A_i, o \rangle$ could be much smaller than 1, forcing $\tau$ to be large. However, we do not need to optimize over these constraints because $\hat{g} \geq 1$, so these constraints do not contribute to the $\max$ in its definition. This leads to the set of non-redundant constraints $I$, which for efficiency reasons, we relax to those indices of constraints $A_i$ that do not cover a box that contains the lens, cf. Lemma C.3. We prove this is good enough in terms of the width parameters. The computation of $I$, which is done before each restart phase, requires $O(N)$ operations and each query to the oracle takes $O(n \log(\frac{n}{(\omega-1)\delta} + \frac{n}{\omega-1}))$ operations, cf. Lemma C.2.

## Acknowledgments and Disclosure of Funding

We thank Elias Koutsoupias for some suggestions and discussions that started the line of research of this work. David Martínez-Rubio and Francisco Criado were partially funded by the DFG Cluster of Excellence MATH+ (EXC-2046/1, project id 390685689) funded by the Deutsche Forschungsgemeinschaft (DFG). David Martínez-Rubio was also partially supported by EP/N509711/1 from the EPSRC MPLS division, grant No 2053152.

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
