## A  Missing proofs from the introduction

**Lemma A.1.** *Recall the definition of the primal problem* (1FP)*:*

$$\max_{x \in \mathbb{R}^n_{\geq 0}} \left\{ f(x) \stackrel{\text{def}}{=} \sum_{i=1}^{n} \log x_i : Ax \leq \mathbb{1}_m \right\}. \tag{1FP}$$

*Then, its Lagrange dual can be formulated as:*

$$\min_{\lambda \in \Delta^m} \left\{ g(\lambda) \stackrel{\text{def}}{=} -\sum_{i=1}^{n} \log(A^T \lambda)_i - n \log n \right\}. \tag{1FP-Dual}$$

*Where* $\Delta^m \stackrel{\text{def}}{=} \{\lambda \in \mathbb{R}^m : \sum \lambda_i = 1, \lambda \geq 0\}$ *is the $m$-dimensional (probability) simplex.*

*Proof.* By definition of Lagrangian duality, the dual of (1FP) is

$$\min_{y \geq 0} \left\{ \max_{x \geq 0} \left\{ \sum_{i=1}^{n} \log(x_i) - y^T(Ax - \mathbb{1}_m) \right\} \right\} =$$
$$\min_{y \geq 0} \left\{ \langle y, \mathbb{1}_m \rangle + \max_{x \geq 0} \left\{ \sum_{i=1}^{n} \left( \log(x_i) - (y^T A)_i x_i \right) \right\} \right\}$$

We can explicitly compute the $x_i$ variables by differentiation, to obtain $x_i = (y^T A)_i^{-1}$. Thus the problem is equivalent to:

$$\min_{y \geq 0} \left\{ \langle y, \mathbb{1}_m \rangle + \sum_{i=1}^{n} \left( -\log(y^T A)_i - 1 \right) \right\}.$$

Now, let $\lambda = \frac{1}{\langle y, \mathbb{1}_m \rangle} y$ and $t = \langle y, \mathbb{1}_m \rangle$. Then, $y = t\lambda$ with $\lambda \in \Delta^m$ and we write the problem as:

$$\min_{\lambda \in \Delta^m} \min_{t \geq 0} \left\{ t - \sum_{i=1}^{n} \log(\lambda^T A)_i - n \log t - n \right\}.$$

The value of $t$ minimizing $t - n \log t$ is $t = n$ by differentiation so the problem reduces to

$$\min_{\lambda \in \Delta^m} \left\{ -\sum_{i=1}^{n} \log(A^T \lambda)_i - n \log n \right\}. \tag{8}$$

$\square$

## B  Missing proofs from Section 2

*Proof* (Proposition 2.1). Recall $\beta = \frac{\varepsilon}{6n \log(2mn^2/\varepsilon)}$ and that $\log(\cdot)$ is the natural logarithm. We will prove the proposition in three steps:

1) $\hat{f}(\hat{x}^*) - \hat{f}(x_r^*) \leq \hat{f}(\hat{x}^*) + f_r(x_r^*) \leq 3\varepsilon$.

2) The point $x_r^\varepsilon$ satisfies $A \exp(x_r^\varepsilon) \leq (1 + \varepsilon/n)\mathbb{1}_m$.

3) The point $\hat{u} = x_r^\varepsilon - \log(1 + \varepsilon/n)\mathbb{1}_n$ satisfies $f(x^*) - f(u) \leq f(\exp(\hat{x}^*)) - f(\exp(\hat{u})) = \hat{f}(\hat{x}^*) - \hat{f}(\hat{u}) \leq 5\varepsilon$ and $A\exp(\hat{u}) \leq \mathbb{1}_m$.

For the first part, take the point $x = \log(1 - \varepsilon/n)\mathbb{1}_n + \hat{x}^* \in B$. It satisfies $A\exp(x) \leq (1 - \varepsilon/n)\mathbb{1}_m$, because $\hat{x}^*$ is feasible. Thus

$$\frac{\beta}{1+\beta}\sum_{i=1}^m (A\exp(x))_i^{\frac{1+\beta}{\beta}} \leq m(1 - \varepsilon/n)^{1/\beta} \leq m\left(\frac{\varepsilon}{2mn^2}\right)^6 \leq \varepsilon. \tag{9}$$

We used $\frac{\beta}{1+\beta} \leq 1$, $\frac{1+\beta}{\beta} \geq \frac{1}{\beta}$, and $(1 - \varepsilon/n)^{\frac{n}{\varepsilon}} \leq e^{-1}$. Consequently, we have

$$\hat{f}(\hat{x}^*) - \hat{f}(x_r^*) \overset{\text{\textcircled{1}}}{\leq} \hat{f}(\hat{x}^*) + f_r(x_r^*) \overset{\text{\textcircled{2}}}{\leq} \hat{f}(\hat{x}^*) + f_r(x)$$

$$= \langle \mathbb{1}_n, \hat{x}^* \rangle + \left( -\langle \mathbb{1}_n, \log(1 - \varepsilon/n)\mathbb{1}_n + \hat{x}^* \rangle + \frac{\beta}{1+\beta}\sum_{i=1}^m (A\exp(x))_i^{\frac{1+\beta}{\beta}} \right) \tag{10}$$

$$\overset{\text{\textcircled{3}}}{\leq} n\log(\frac{1}{1-\varepsilon/n}) + \varepsilon \overset{\text{\textcircled{4}}}{\leq} 3\varepsilon.$$

Above, \textcircled{1} is true by definition of $f_r$ being $-\hat{f}$ plus a non-negative regularizer. The point $x$ is in $B$ and $x_r^* = \arg\min_{x \in B}\{f_r(x)\}$ so we have \textcircled{2}. Inequality \textcircled{3} uses (9) and \textcircled{4} uses $\log(x) \leq x - 1$ and $\varepsilon/n \leq 1/2$.

For the second part, suppose for the moment that there is some $i$ such that $(A\exp(x_r^\varepsilon))_i > 1 + \varepsilon/n$. In that case

$$(A\exp(x_r^\varepsilon))_i^{\frac{1+\beta}{\beta}} \geq (1 + \varepsilon/n)^{(2n/\varepsilon)\cdot 3\log(2mn^2/\varepsilon)} \geq \left(\frac{2mn^2}{\varepsilon}\right)^3,$$

since $(1 + \varepsilon/n)^{2n/\varepsilon} \geq e$ when $\varepsilon/n \leq 1/2$. We have $x_r^\varepsilon \in B$ so $f_r(x_r^\varepsilon) \geq -\langle \mathbb{1}_n, x_r^\varepsilon \rangle + \frac{\beta}{1+\beta}\left(\frac{2mn^2}{\varepsilon}\right)^3 \geq \frac{\beta}{2}\left(\frac{2mn^2}{\varepsilon}\right)^3$. On the other hand, it holds for the point $y = -\log(mn)\mathbb{1}_n$ that

$$f_r(y) = n\log(mn) + \frac{\beta}{1+\beta}\sum_{i=1}^m (A\exp(y))_i^{\frac{1+\beta}{\beta}}$$

$$\overset{\text{\textcircled{1}}}{\leq} n\log(mn) + m\,(1/m)^{\frac{1+\beta}{\beta}} \leq n\log(mn) + 1 \tag{11}$$

$$\overset{\text{\textcircled{2}}}{<} \frac{\beta}{2}\left(\frac{2mn^2}{\varepsilon}\right)^3 - \varepsilon < f_r(x_r^\varepsilon) - \varepsilon,$$

contradicting the assumption $f_r(x_r^\varepsilon) - f_r(x_r^*) \leq \varepsilon$, as we would obtain $\varepsilon < f_r(x_r^\varepsilon) - f_r(y) \leq f_r(x_r^\varepsilon) - f_r(x_r^*)$, since $y \in B$. So it must be $(A\exp(x_r^\varepsilon))_i \leq 1 + \varepsilon/n$. Inequality \textcircled{1} uses that the maximum entry of $A$ is 1, and $\frac{\beta}{1+\beta} \leq 1$. One can show \textcircled{2} by proving the stronger inequality that results from substituting $\beta$ by $\varepsilon/(6n \cdot 2mn^2/\varepsilon)$, which is a lower value. Computing derivatives in both sides shows that this inequality holds if it does for $m = 1$ and $\varepsilon = \frac{n}{2}$, and the latter is easy to check.

For the third part, we have $A\exp(\hat{u}) = A\frac{\exp(x_r^\varepsilon)}{1+\varepsilon/n} \leq \mathbb{1}_m$. And finally, putting all together we obtain

$$\begin{aligned}
\hat{f}(\hat{x}^*) - \hat{f}(\hat{u}) &= \hat{f}(\hat{x}^*) - \hat{f}(x_r^\varepsilon) + n\log(1 + \varepsilon/n) \\
&\leq \hat{f}(\hat{x}^*) + f_r(x_r^\varepsilon) + n\log(1 + \varepsilon/n) \\
&\leq \hat{f}(\hat{x}^*) + f_r(x_r^*) + \varepsilon + n\log(1 + \varepsilon/n) \\
&\leq 4\varepsilon + n\log(1 + \varepsilon/n) \leq 5\varepsilon.
\end{aligned} \tag{12}$$

$\square$

**Lemma B.1.** *Let $A$ satisfy the normalization in* (1)*, and let $x^*$ be the optimizer of Problem* (1FP)*. Then $x_i^* \geq 1/n$, for all $i \in [n]$.*

*Proof.* The normalization ensures that $e_i$ are feasible points, for $i \in [n]$. That is, $Ae_i \leq \mathbb{1}_m$ because each $A_{ij} \leq 1$. Since $x^*$ is the maximizer of Problem (1FP), by the first order optimality condition we have $\langle \nabla f(x^*), x - x^* \rangle \leq 0$, for any feasible point $x$. Suppose there is a coordinate $i \in [n]$ such that $x_i^* < \frac{1}{n}$. Then, $\langle \nabla f(x^*), e_i - x^* \rangle = \frac{1}{x_i^*} - \sum_{j=1}^n x_j^*/x_j^* > 0$, which is a contradiction. $\qquad \square$

**Lemma B.2.** *The iterates of Algorithm 1 remain in the box $B$.*

*Proof.* For all $k \geq 0$, we have $z^{(k)} \in B$ by definition. If we have that $y^{(k-1)} \in B$, then $x^{(k)} \in B$ since $x^{(k)}$ is a convex combination of $y^{(k-1)}$ and $z^{(k-1)}$. So we only have to prove that for all $k \geq 0$, we have $y^{(k)} \in B$. We prove by induction that, for $k \geq 1$, it holds that $y^{(k)}$ is a convex combination of $\{z^{(i)}\}_{i=0}^k$ and that the weight of $z^{(k)}$ in this convex combination is $\frac{1}{\eta_k L}$. Firstly, we have $y^{(1)} = (1 - \frac{1}{\eta_1 L})z^{(0)} + \frac{1}{\eta_1 L} z^{(1)}$ (recall $x^{(0)} = z^{(0)}$). Now assuming our property holds up to $k - 1$, use the definition of $y^{(k)}$ and $x^{(k)}$, to compute $y^{(k)} = \tau z^{(k-1)} + (1 - \tau)y^{(k-1)} + \frac{1}{\eta_k L}(z^{(k)} - z^{(k-1)})$. This is an affine combination of the $z^{(i)}$'s, by induction hypothesis. Moreover, the weights add up to $1 = \tau + (1 - \tau) + \frac{1}{\eta_k L} - \frac{1}{\eta_k L}$, and the weight on $z^{(k)}$ is $\frac{1}{\eta_k L}$. So we only have to prove the weight on $z^{(k-1)}$ is $\geq 0$ in order to show that we indeed have a convex combination and not just an affine one. By induction hypothesis, we know the weight on $z^{(k-1)}$ coming from $y^{(k-1)}$ is $\frac{1}{\eta_{k-1} L}$. Hence, the weight on $z^{(k-1)}$ is $\tau + (1 - \tau)\frac{1}{\eta_{k-1} L} - \frac{1}{\eta_k L} = \tau > 0$, where the equality uses the definition of $\eta_k$. $\qquad \square$

*Proof* (Lemma 2.2). Recall that we want to prove that for the parameters in Algorithm 1 and for $u \in B$ we have

$$\langle \eta_k \overline{\nabla f_r}(x^{(k)}), z^{(k-1)} - u \rangle \leq \eta_k^2 L \langle \overline{\nabla f_r}(x^{(k)}), x^{(k)} - y^{(k)} \rangle + \frac{1}{2\omega}\|z^{(k-1)} - u\|_2^2 - \frac{1}{2\omega}\|z^{(k)} - u\|_2^2.$$

Use Lemma B.3.b) below with loss $\ell^{(k)} = \overline{\nabla f_r}(x^{(k)})$, learning rate $\eta = \eta_k$, and regularizer $\psi(x) = \frac{1}{2\omega}\|x\|_2^2$, that yields Bregman divergence $D_\psi(x, y) = \frac{1}{2\omega}\|x - y\|_2^2$. Use that $z^{(k-1)} - z^{(k)} = \eta_k L(x^{(k)} - y^{(k)})$. $\qquad \square$

**Lemma B.3 (Mirror Descent Lemma).** *Let $\mathcal{X} \subseteq \mathbb{R}^n$ be a closed convex set and let $\psi : \mathcal{X} \to \mathbb{R}$ be a 1-strongly convex map with respect to $\|\cdot\|$. Let $\|\cdot\|_*$ be the dual norm to $\|\cdot\|$ and let $\ell^{(k)} \in \mathbb{R}^n$ be an arbitrary loss vector. Given $z^{(k-1)} \in \mathcal{X}$, let $z^{(k)} \overset{\text{def}}{=} \arg\min_{z \in \mathcal{X}}\{D_\psi(z, z^{(k-1)}) + \eta\langle \ell^{(k)}, z \rangle\}$. Then, for all $u \in \mathcal{X}$ we have*

*a)* $\eta\langle \ell^{(k)}, z^{(k-1)} - u \rangle \leq \frac{\eta^2}{2}\|\ell^{(k)}\|_*^2 + D_\psi(u, z^{(k-1)}) - D_\psi(u, z^{(k)})$.

*b)* $\eta\langle \ell^{(k)}, z^{(k-1)} - u \rangle \leq \eta\langle \ell^{(k)}, z^{(k-1)} - z^{(k)} \rangle + D_\psi(u, z^{(k-1)}) - D_\psi(u, z^{(k)})$.

*Proof.* We note that, by definition, we have $\frac{\partial}{\partial x}D_\psi(x, y) = \nabla\psi(x) - \nabla\psi(y)$. The lemma is due to

$$\langle \eta\ell^{(k)}, z^{(k-1)} - u \rangle = \langle \eta\ell^{(k)}, z^{(k-1)} - z^{(k)} \rangle + \langle \eta\ell^{(k)}, z^{(k)} - u \rangle$$

$$\overset{①}{\leq} \langle \eta\ell^{(k)}, z^{(k-1)} - z^{(k)} \rangle - \langle \nabla\psi(z^{(k)}) - \nabla\psi(z^{(k-1)}), z^{(k)} - u \rangle$$

$$\overset{②}{=} \langle \eta\ell^{(k)}, z^{(k-1)} - z^{(k)} \rangle - D_\psi(z^{(k)}, z^{(k-1)}) + D_\psi(u, z^{(k-1)}) - D_\psi(u, z^{(k)})$$

$$\overset{③}{\leq} \frac{\eta^2}{2}\|\ell^{(k)}\|_*^2 + D_\psi(u, z^{(k-1)}) - D_\psi(u, z^{(k)}).$$

Inequality ① comes from the first-order optimality condition of the definition of $z^{(k)}$, that is, $\langle \nabla\psi(z^{(k)}) - \nabla\psi(z^{(k-1)}) + \eta\ell^{(k)}, u - z^{(k)} \rangle \geq 0$ for all $u \in \mathcal{X}$. ② is the triangle equality of Bregman divergences, and can be easily checked by using the definition.

If we drop the term $-D_\psi(z^{(k)}, z^{(k-1)})$ after ②, we obtain part b) of this lemma. ③ leads to part a), which is the classical mirror descent lemma. It uses the bound $D_\psi(z^{(k)}, z^{(k-1)}) \geq \frac{1}{2}\|z^{(k)} - z^{(k-1)}\|^2$, which holds due to the strong convexity of $\psi$. And then we applied the inequality $\langle v, w \rangle - \frac{1}{2}\|w\|^2 \leq \frac{1}{2}\|v\|_*^2$ for $v, w \in \mathbb{R}^n$, that holds by Cauchy-Schwarz and $\|v\|_* \cdot \|w\| \leq \frac{1}{2}\|v\|_*^2 + \frac{1}{2}\|w\|^2$.

$\qquad \square$

We now prove the descent step Lemma 2.3. In the proof, we will use Lemma B.4 below, which is a crucial generalization of [DFO20, Lemma 3.1]. We first prove Lemma 2.3.

*Proof* (Lemma 2.3). We have $x^{(k)} - y^{(k)} = (z^{(k-1)} - z^{(k)})/\eta_k L$ by definition of the gradient descent step. With this, we first conclude that $\frac{1}{2}\langle \nabla f_r(x^{(k)}), x^{(k)} - y^{(k)} \rangle \geq 0$, as $\nabla_i f_r(x^{(k)})$ and $x_i^{(k)} - y_i^{(k)}$ have the same sign for all $i \in [n]$, cf. (4).

We apply Lemma B.4 with $y^{(k)}$ corresponding to $x + \Delta$ and $x^{(k)}$ corresponding to $x$. To this end, we choose $c_i \geq 0$ satisfying ① below

$$\frac{c_i \beta}{4(1+\beta)}|\overline{\nabla_i f_r}(x^{(k)})| \overset{①}{=} |x_i^{(k)} - y_i^{(k)}| \overset{②}{=} \frac{1}{\eta_k L}|z_i^{(k-1)} - z_i^{(k)}| \overset{③}{\leq} \frac{\omega}{L}|\overline{\nabla_i f_r}(x^{(k)})|,$$

where ② holds by definition of $y^{(k)}$ and ③ holds by the mirror descent update (4). Thus, it suffices to pick $c_i$ such that $c_i \leq \frac{4\omega(1+\beta)}{\beta L} \leq 1$, where the last inequality holds true by the definition of $L$. In fact, the value of $L$ was chosen to satisfy the previous inequality. Hence, Lemma B.4 can be applied. We obtain:

$$f_r(x^{(k)}) - f_r(y^{(k)}) \geq \sum_{i=1}^{n} \left(1 - \frac{c_i}{2}\right) \nabla_i f_r(x^{(k)})(x_i^{(k)} - y_i^{(k)}) \geq \frac{1}{2}\langle \nabla f_r(x^{(k)}), x^{(k)} - y^{(k)} \rangle.$$

as desired. $\qquad\qquad\square$

**Lemma B.4.** *Let $c \in [-1,1]^n$ and let $\Delta \in \mathbb{R}^n$ be defined as $\Delta_j = -\frac{c_j \beta}{4(1+\beta)}\overline{\nabla_j f_r}(x)$, for $j \in [n]$. Then*

$$f_r(x + \Delta) - f_r(x) \leq \sum_{j=1}^{n}(1 - \frac{c_j}{2})\Delta_j \nabla_j f_r(x).$$

*Proof.* By using a Taylor expansion, there is a $t \in [0,1]$ such that

$$f_r(x + \Delta) - f_r(x) = \langle \nabla f_r(x), \Delta \rangle + \frac{1}{2}\Delta^\top \nabla^2 f_r(x + t\Delta)\Delta. \tag{13}$$

The gradient and Hessian of $f_r$ are given by

$$\nabla_j f_r(x) = -1 + \sum_{i=1}^{m}(A\exp(x))_i^{\frac{1}{\beta}} a_{ij}\exp(x_j),$$

$$\nabla_{jk}^2 f_r(x) = \mathbb{1}_{\{j=k\}}\sum_{i=1}^{m}(A\exp(x))_i^{\frac{1}{\beta}} a_{ij}\exp(x_j) \tag{14}$$

$$+ \frac{1}{\beta}\sum_{i=1}^{m}(A\exp(x))_i^{\frac{1}{\beta}-1} a_{ij}\exp(x_j)a_{ik}\exp(x_k).$$

In order to control how much the function changes, we will require

$$\frac{1}{2}\nabla_{jk}^2 f_r(x) \leq \nabla_{jk}^2 f_r(x + t\Delta) \leq 2\nabla_{jk}^2 f_r(x).$$

We can guarantee the inequality on the right if we guarantee that each summand in the expression above does not grow by more than a factor of 2, and respectively the one on the left if it does not decrease by more than a factor of 2. Let $\Delta_{\max} = \max_{i\in[n]}\{\Delta_i\}$ and $\Delta_{\min} = \min_{i\in[n]}\{\Delta_i\}$. It suffices to have $\exp(\Delta_{\max})^{\frac{1}{\beta}+1} \leq 2$ and $\exp(\Delta_{\min})^{\frac{1}{\beta}+1} \geq 1/2$. Hence, it suffices to have for all $j \in [n]$, the following:

$$-\frac{\ln 2}{1 + \frac{1}{\beta}} \leq \Delta_j \leq \frac{\ln 2}{1 + \frac{1}{\beta}}. \tag{15}$$

In fact, we will use $\Delta_j = -\frac{c_j}{4} \cdot \frac{\beta}{1+\beta}\overline{\nabla_j f_r}(x)$ for all $j \in [n]$, which satisfy the condition since $|c_j\overline{\nabla_j f_r}(x)| \leq 1$. In such a case, by using the function $\text{sign}(x) = 1$ if $x \geq 0$ and $-1$ otherwise, we

have:

$$\frac{1}{2}\Delta^\top \nabla^2 f_r(x+t\Delta)\Delta \overset{\textcircled{1}}{\le} \sum_{j=1}^{n}\sum_{i=1}^{m}(A\exp(x))_i^{\frac{1}{\beta}}\Delta_j^2 a_{ij}\exp(x_j)$$

$$+\frac{1}{2\beta}\sum_{i=1}^{m}(A\exp(x))_i^{\frac{1}{\beta}-1}\sum_{j=1}^{n}\sum_{k=1}^{n}\Delta_j\Delta_k a_{ij}a_{ik}\exp(x_j)\exp(x_j)2^{\mathrm{sign}(\Delta_j\Delta_k)}$$

$$\overset{\textcircled{2}}{\le}\sum_{j=1}^{n}\sum_{i=1}^{m}(A\exp(x))_i^{\frac{1}{\beta}}\Delta_j^2 a_{ij}\exp(x_j)$$

$$+\frac{1}{2\beta}\sum_{i=1}^{m}(A\exp(x))_i^{\frac{1}{\beta}-1}\sum_{j=1}^{n}\Delta_j^2 a_{ij}\exp(x_j)\cdot\sqrt{\sum_{j=1}^{n}\sum_{k=1}^{n}a_{ij}a_{ik}\exp(x_j)\exp(x_k)4^{\mathrm{sign}(\Delta_j\Delta_k)}}$$

$$\overset{\textcircled{3}}{\le}\frac{\beta+1}{\beta}\sum_{j=1}^{n}\sum_{i=1}^{m}(A\exp(x))_i^{\frac{1}{\beta}}\Delta_j^2 a_{ij}\exp(x_j)$$

$$\overset{\textcircled{4}}{=}\frac{\beta+1}{\beta}\sum_{j=1}^{n}\Delta_j^2(\nabla_j f_r(x)+1)$$

$$\overset{\textcircled{5}}{=}\sum_{j=1}^{n}-\frac{c_j}{4}\Delta_j\overline{\nabla_j f_r}(x)(\nabla_j f_r(x)+1)$$

$$\overset{\textcircled{6}}{\le}-\sum_{j=1}^{n}\frac{c_j}{2}\Delta_j\nabla_j f_r(x).$$

$$(16)$$

We used the inequalities $\nabla_{jk}^2 f_r(x+t\Delta)\ \le\ 2\nabla_{jk}^2 f_r(x)$ and $-\nabla_{jk}^2 f_r(x+t\Delta)\ \le\ -2^{-1}\nabla_{jk}^2 f_r(x)$ in $\textcircled{1}$. We used Cauchy-Schwarz in $\textcircled{2}$ with the $n^2$-dimensional vectors $(\Delta_j\sqrt{a_{ij}a_{ik}\exp(x_j)\exp(x_k)})_{j,k\in[n]}$ and $(2^{\mathrm{sign}(\Delta_j\Delta_k)}\sqrt{a_{ij}a_{ik}\exp(x_j)\exp(x_k)})_{j,k\in[n]}$ in order to bound the last factor, so that the two first lines of the right hand side become proportional after bounding $4^{\mathrm{sign}(\Delta_j\Delta_k)}\le 4$ in $\textcircled{3}$. In $\textcircled{3}$, we also grouped these terms. In $\textcircled{4}$, we used the definition of the gradient. In $\textcircled{5}$, we used the value of $\Delta$. Finally, $\textcircled{6}$ is a direct consequence of the truncated gradient definition (one can check the inequality for the three cases in $\nabla_j f_r(x)\in\{[-1,0),[0,1],(1,\infty)\}$, while taking into account the sign of $\Delta_j$).

Now, substituting into (13) we obtain:

$$f_r(x+\Delta)-f_r(x)\le\sum_{j=1}^{n}\left(1-\frac{c_j}{2}\right)\Delta_j\nabla_j f_r(x).$$

$\square$

*Proof* (Lemma 2.4). It is enough to show that for all $i\in[n]$ we have

$$\eta_k\nu_i^{(k)}(z_i^{(k-1)}-u_i)+\eta_k^2 L\overline{\nabla_i f_r}(x^{(k)})(x_i^{(k)}-y_i^{(k)})\le\frac{3}{2}\eta_k L\nabla_i f_r(x^{(k)})(x_i^{(k)}-y_i^{(k)}) \qquad (17)$$

because then we can conclude with

$$\langle\eta_k\nu^{(k)},z^{(k-1)}-u\rangle+\eta_k^2 L\langle\overline{\nabla f_r}(x^{(k)}),x^{(k)}-y^{(k)}\rangle\overset{\textcircled{1}}{\le}\frac{3}{2}\eta_k L\langle\nabla f_r(x^{(k)}),x^{(k)}-y^{(k)}\rangle$$
$$\overset{\textcircled{2}}{\le}3\eta_k L(f_r(x^{(k)})-f_r(y^{(k)})),$$

$$(18)$$

by adding up (17) in $\textcircled{1}$ and Lemma 2.3 in $\textcircled{2}$. In the analysis of (17) we exploit the simple but crucial fact that is that the gradient step for each coordinate is independent of the gradient step of other coordinates, due to the constraint set being a box. We present the rest of the proof in three cases. In the cases below, we will use $\nabla_i f_r(x^{(k)})(x_i^{(k)}-y_i^{(k)})\ge 0$, cf. Lemma 2.3. And also the fact that $\eta_k\le 1/4$, as we observe in (22).

- If $\nu_i^{(k)} = 0$ then $\overline{\nabla_i f_r}(x^{(k)}) = \nabla_i f_r(x^{(k)}) \in [-1, 1]$. In such a case, we have

$$\eta_k \nu_i^{(k)}(z_i^{(k-1)} - u_i) + \eta_k^2 L \overline{\nabla_i f_r}(x^{(k)})(x_i^{(k)} - y_i^{(k)}) = \eta_k^2 L \nabla_i f_r(x^{(k)})(x_i^{(k)} - y_i^{(k)})$$
$$\leq \frac{3}{2} \eta_k L \nabla_i f_r(x^{(k)})(x_i^{(k)} - y_i^{(k)}).$$

- If $\nu_i^{(k)} > 0$ and $z_i^{(k)} > -\omega$ then the mirror descent step did not need to project along coordinate $i$, and we have $z_i^{(k)} = z_i^{(k-1)} - \omega \eta_k$, and thus $y_i^{(k)} = x_i^{(k)} - \omega/L$. In this case

$$\eta_k \nu_i^{(k)}(z_i^{(k-1)} - u_i) + \eta_k^2 L \overline{\nabla_i f_r}(x^{(k)})(x_i^{(k)} - y_i^{(k)})$$
$$\overset{\textcircled{1}}{\leq} \eta_k \nabla_i f_r(x^{(k)})\omega + \eta_k^2 L \nabla_i f_r(x^{(k)})(x^{(i)} - y^{(i)})$$
$$= \eta_k L \nabla_i f_r(x^{(k)})(x_i^{(k)} - y_i^{(k)}) + \eta_k^2 L \nabla_i f_r(x^{(k)})(x_i^{(k)} - y_i^{(k)})$$
$$\leq \frac{3}{2} \eta_k L \nabla_i f_r(x^{(k)})(x_i^{(k)} - y_i^{(k)}).$$

Above, we obtain $\textcircled{1}$ from $z_i^{(k)} - u_i \leq \omega$ because $z^{(k)}, u \in B$, the fact that $\nu_i^{(k)}$ and $x_i^{(k)} - y_i^{(k)}$ are positive, and $1 = \overline{\nabla_i f_r}(x^{(k)}) \leq \nabla_i f_r(x^{(k)}), 0 < \nu_i^{(k)} \leq \nabla_i f_r(x^{(k)})$.

- If $\nu_i^{(k)} > 0$ and $z_i^{(k)} = -\omega$ then

$$\eta_k \nu_i^{(k)}(z_i^{(k-1)} - u_i) + \eta_k^2 L \overline{\nabla_i f_r}(x^{(k)})(x_i^{(k)} - y_i^{(k)})$$
$$\overset{\textcircled{1}}{\leq} \eta_k \nabla_i f_r(x^{(k)})(z_i^{(k-1)} - z_i^{(k)}) + \eta_k^2 L \nabla_i f_r(x^{(k)})(x_i^{(k)} - y_i^{(k)})$$
$$\overset{\textcircled{2}}{\leq} \eta_k^2 L \nabla_i f_r(x^{(k)})(x_i^{(k)} - y_i^{(k)}) + \eta_k^2 L \nabla_i f_r(x^{(k)})(x_i^{(k)} - y_i^{(k)})$$
$$\leq \frac{3}{2} \eta_k L \nabla_i f_r(x^{(k)})(x_i^{(k)} - y_i^{(k)}).$$

We have $\textcircled{1}$ because in this case, $u_i - z_i^{(k)}$, $x_i^{(k)} - y_i^{(k)}$, $\nu_i^{(k)}$, $\overline{\nabla_i f_r}(x^{(k)})$ are all $\geq 0$. We also used $0 < \nu_i^{(k)} < \nabla_i f_r(x^{(k)}), 0 < \overline{\nabla_i f_r}(x^{(k)}) < \nabla_i f_r(x^{(k)})$. In $\textcircled{2}$, we used $z_i^{(k-1)} - z_i^{(k)} = \eta_k L(x_i^{(k)} - y_i^{(k)})$. $\qquad\square$

*Proof* ([Theorem 2.5](#)). We start by bounding the gap with respect to $x^{(k)}$:

$$\eta_k(f_r(x^{(k)}) - f_r(u))$$
$$\overset{\textcircled{1}}{\leq} \langle \eta_k \nabla f_r(x^{(k)}), x^{(k)} - u \rangle$$
$$= \langle \eta_k \nabla f_r(x^{(k)}), x^{(k)} - z^{(k-1)} \rangle + \langle \eta_k \nu^{(k)}, z^{(k-1)} - u \rangle + \langle \eta_k \overline{\nabla f_r}(x^{(k)}), z^{(k-1)} - u \rangle$$
$$\overset{\textcircled{2}}{=} \frac{(1-\tau)\eta_k}{\tau} \langle \nabla f_r(x^{(k)}), y^{(k-1)} - x^{(k)} \rangle + \langle \eta_k \nu^{(k)}, z^{(k-1)} - u \rangle + \langle \eta_k \overline{\nabla f_r}(x^{(k)}), z^{(k-1)} - u \rangle$$
$$\overset{\textcircled{3}}{\leq} \frac{(1-\tau)\eta_k}{\tau}(f_r(y^{(k-1)}) - f_r(x^{(k)})) + \langle \eta_k \nu^{(k)}, z^{(k-1)} - u \rangle + \langle \eta_k^2 L \overline{\nabla f_r}(x^{(k)}), x^{(k)} - y^{(k)} \rangle$$
$$+ \frac{1}{2\omega}\|z^{(k-1)} - u\|_2^2 - \frac{1}{2\omega}\|z^{(k)} - u\|_2^2]$$
$$\overset{\textcircled{4}}{\leq} \frac{(1-\tau)\eta_k}{\tau}(f_r(y^{(k-1)}) - f_r(x^{(k)})) + C_k(f_r(x^{(k)}) - f_r(y^{(k)})) + \frac{1}{2\omega}\|z^{(k-1)} - u\|_2^2$$
$$- \frac{1}{2\omega}\|z^{(k)} - u\|_2^2$$
$$\overset{\textcircled{5}}{\leq} \eta_k f_r(x^{(k)}) + (C_k - \eta_k)f_r(y^{(k-1)}) - C_k f_r(y^{(k)}) + \frac{1}{2\omega}\|z^{(k-1)} - u\|_2^2 - \frac{1}{2\omega}\|z^{(k)} - u\|_2^2$$

$$(19)$$

We used convexity in ①. The definition of $x^{(k)}$ is used in ②. Inequality ③ uses convexity and Lemma 2.2. We applied Lemma 2.4 in ④. In ⑤, we substituted the value of $\tau$, which is picked to be $\tau \stackrel{\text{def}}{=} \eta_k/C_k = \frac{1}{3L}$ so we can cancel $f_r(x^{(k)})$ in both sides of (19).

The choice of $\eta_k$ is made so that $C_k - \eta_k = C_{k-1}$ (or equiv. $(3L-1)\eta_k = 3L\eta_{k-1}$), which allows to telescope the previous expression. Adding up (19) for $k = 1, \ldots, T$ with $u = x_r^*$, we have

$$\left(-C_0 - \sum_{k=1}^{T} \eta_k\right) f_r(x_r^*) \leq C_0(f_r(y^{(0)}) - f_r(x_r^*)) - C_T f_r(y^{(T)}) + \frac{1}{2\omega}\|z^{(0)} - x_r^*\|_2^2.$$

We dropped $-\frac{1}{2\omega}\|z^{(T)} - x_r^*\|_2^2 \leq 0$. Now, since $\eta_k = C_k - C_{k-1}$ we have $-C_0 - \sum_{k=1}^{T} \eta_k = -C_T$. So reorganizing terms we obtain

$$f_r(y^{(T)}) \leq f_r(x_r^*) + \frac{1}{C_T}\left(C_0(f_r(y^{(0)}) - f_r(x_r^*)) + \frac{1}{2\omega}\|z^{(0)} - x_r^*\|_2^2\right)$$

$$\stackrel{①}{\leq} f_r(x_r^*) + \frac{1}{C_T}\left(C_0(n(\log(2mn) + 1) + \frac{n\log(mn)}{2}\right) \tag{20}$$

$$\stackrel{②}{\leq} f_r(x_r^*) + \varepsilon$$

Above, ① uses $f_r(y^{(0)}) \leq n\log(mn/(1 - \varepsilon/n)) + \varepsilon \leq n(\log(2mn) + 1)$ and $-f_r(x_r^*) \leq 0$. For the former, take into account that $-\log(mn)\mathbb{1}_n$ is feasible and so the regularizer at $y^{(0)} = -\log(mn/(1 - \varepsilon/n))\mathbb{1}_n$ is at most $\varepsilon$, cf. (9). Recall $\varepsilon < n/2$. We also bounded the last summand by using that $z^{(0)}, x_r^* \in B$ so $\|z^{(0)} - x_r^*\|_2^2 \leq n\omega^2$.

At this point, the only free parameters left are $C_0$ (via $\eta_0$) and $T$. We set $\eta_0 = \frac{1}{3L}$ so that $C_0 = 1$. And we have that $C_T = 3L\eta_T = 3L\eta_0(1 - \tau)^{-T} = (1 - \tau)^{-T}$. So if we pick $T$ such that

$$\frac{1}{C_T} = (1 - \tau)^T \leq \frac{\varepsilon}{4n\log(2mn)}, \tag{21}$$

we will obtain ②. We will pick the smallest $T$ that satisfies (21). That is,

$$T = \left\lceil \frac{\log(4n\log(2mn)/\varepsilon)}{\log(1/(1-\tau))} \right\rceil \leq \left\lceil 3L\log\left(\frac{4n\log(2mn)}{\varepsilon}\right) \right\rceil = \widetilde{O}(n/\varepsilon).$$

On the other hand, by definition of $T$ as the minimum natural number satisfying (21), we have,

$$(1 - \tau)^T = \frac{\eta_0}{\eta_T} \geq \frac{\varepsilon}{4n\log(2mn)}(1 - \tau).$$

We can use this inequality to show $\eta_k \leq \frac{1}{4}$, for all $k \in [T]$, which is used in the proof of Lemma 2.4. It is enough that ① below is satisfied:

$$\eta_k \leq \eta_T \leq \frac{4n\log(2mn)\eta_0}{\varepsilon}\frac{1}{1 - \tau} = \frac{4n\log(2mn)}{3L\varepsilon}\frac{3L}{3L - 1} \stackrel{①}{\leq} \frac{1}{4}. \tag{22}$$

If $L \geq \frac{16n\log(2mn)}{3\varepsilon} + \frac{1}{3}$ then ① holds, and we chose $L$ to satisfy this inequality.

We note we could increase $T$ by a factor $C > 1$ inside of its $\log$ in the numerator, so that the error obtained in ② is $\varepsilon/C$. However, the requirement on $L$ above would increase by a factor of $C$, so we would end up with an extra factor of $C$ in the value of $T$. Also, the reduction caused by the smoothing already incurs in an $\varepsilon$ additive error, cf. Proposition 2.1. Part 1 of Proposition 2.1 could use $x = \log(1 - \frac{\varepsilon}{nC})\mathbb{1}_n + \hat{x}^*$ so that inequality (10) ends up being bounded by $O(\varepsilon/C)$, but that would require to have $\beta$ be $C$ times smaller for (9) to work. This would also require to make $L$ larger by a factor of $C$ so we would also end up having the $C$ in the total number of iterations to obtain an $O(\varepsilon)$-optimizer.

In conclusion, we obtain an $\varepsilon$ minimizer of $f_r$ in $\widetilde{O}(n/\varepsilon)$ iterations and by Proposition 2.1, we get that $\bar{x}$ is a feasible point that is a $5\varepsilon$ optimizer of Problem (1FP). Finally, we note that each iteration of the algorithm can be implemented in $O(N)$ operations, that are distributed, where the bottleneck is the computation of the gradient. From the definition of the gradient, it is clear that it can be computed in our distributed model of computation and that each agent only needs their local variables for the rest of the steps in the algorithm. □

# C Missing proofs from Section 3

*Proof* (Lemma 3.1). As $Ap^\lambda \leq (1+\varepsilon)\mathbb{1}_m$ we have $A\frac{p^\lambda}{1+\varepsilon} \leq \mathbb{1}_m$ and hence $\frac{p^\lambda}{1+\varepsilon}$ is primal feasible. Therefore,

$$
\begin{aligned}
g(\lambda) - g(\lambda^*) &\leq g(\lambda) - f\left(\frac{p^\lambda}{1+\varepsilon}\right) \\
&= -\sum_{i\in[n]} \log(A^T\lambda)_i - n\log n - \sum_{i\in[n]} \log\left(\frac{1}{n(1+\varepsilon)(A^T\lambda)_i}\right) \\
&= n\log(1+\varepsilon) \leq n\varepsilon.
\end{aligned}
\tag{23}
$$

$\square$

*Proof* (Lemma 3.2). Let us look at one coordinate $i \in [n]$. By the weighted harmonic-arithmetic inequality:

$$
\begin{aligned}
c(\zeta_1 c(p^{(1)}) + \cdots + \zeta_k c(p^{(k)}))_i &= \left(\zeta_1\frac{1}{p_i^{(1)}} + \cdots + \zeta_k\frac{1}{p_i^{(k)}}\right)^{-1} \\
&\leq \zeta_1 p_i^{(1)} + \cdots + \zeta_k p_i^{(k)} = \left(\zeta_1 p^{(1)} + \cdots + \zeta_k p^{(k)}\right)_i.
\end{aligned}
\tag{24}
$$

Now we prove convexity of $c(\mathcal{D}^+)$. Let $q^{(1)}, \ldots, q^{(k)} \in c(\mathcal{D}^+)$. Since every constraint in $\mathcal{D}^+$ is coordinate-wise smaller than some constraint in $\mathcal{D}$, it follows that every point $q^{(j)} \in c(\mathcal{D}^+)$ is coordinate-wise larger than some point $p^{(j)} \in c(\mathcal{D})$, that is $q^{(j)} \geq p^{(j)}$ for $j \in [k]$. Thus we obtain

$$
\sum_{j\in[k]} \zeta_j q^{(j)} \geq \sum_{j\in[k]} \zeta_j p^{(j)} \overset{①}{\geq} c\left(\sum_{j\in[k]} \zeta_j c(p^{(j)})\right) \overset{②}{\in} c(\mathcal{D}).
$$

Inequality ① is the first part of the lemma above, and the membership ② is due to the convexity of $\mathcal{D}$, which is a polytope. The constraints in $c(\mathcal{D}^+)$ are exactly the ones that are coordinate-wise greater than some constraint in $c(\mathcal{D})$, so we have $\sum_{j\in[k]} \zeta_j q_j \in c(\mathcal{D}^+)$. $\square$

**Lemma C.1** (Multiplicative Weights Lemma - Additive and Mult. Guarantee). *Let $\ell^{(k)} \in [-\sigma,\tau]^{\widetilde{m}}$ be an sequence of $\widetilde{m}$-dimensional arbitrary loss vectors, for $k \in [K]$ and $\sigma, \tau \in \mathbb{R}_{>0}$. Denote $W^+ = \max\{\sigma,\tau\}, W^- = \min\{\sigma,\tau\}$. For a target accuracy $\delta \in (0, 2W^-]$, learning rate $\eta = \frac{\delta}{4W^-} \leq \frac{1}{2}$, and initial weights $\Lambda^{(1)} = \mathbb{1}_{\widetilde{m}} \in \Delta^{\widetilde{m}}$, inducing an initial uniform distribution, run the following multiplicative weights update rule*

$$
\Lambda^{(k+1)} \leftarrow \Lambda^{(k)} \odot (\mathbb{1}_{\widetilde{m}} - \frac{\eta}{W^+}\ell^{(k)}),
$$

*for $k = 1, \ldots, K \overset{\text{def}}{=} \frac{8\sigma\tau\log(\widetilde{m})}{\delta^2}$, where $\odot$ represents the coordinate-wise product. Then, for every $u \in \Delta^{\widetilde{m}}$ we have*

$$
\frac{1}{K}\sum_{k=1}^{K}\langle \ell^{(k)}, \frac{\Lambda^{(k)}}{\|\Lambda^{(k)}\|_1}\rangle \leq \delta + \frac{1 + \text{sign}(\sigma,\tau)\eta}{K}\sum_{k=1}^{K}\langle \ell^{(k)}, u\rangle,
$$

*where $\text{sign}(\sigma,\tau)$ is 1 if $\tau \geq \sigma$ and $-1$ otherwise.*

*Proof.* We assume without loss of generality that for a fixed $i \in [\widetilde{m}]$ we have $u = e_i$. It is enough to prove the result in this case since the general case can be obtained as a convex combination of the resulting inequalities.

We use the potential function $\Phi^{(k)} \stackrel{\text{def}}{=} \|\Lambda^{(k)}\|_1$. On the one hand we have

$$\Phi^{(K+1)} \stackrel{\text{①}}{=} \sum_{i=1}^{\widetilde{m}} \Lambda_i^{(K)} \left(1 - \frac{\eta}{W^+}\ell_i^{(K)}\right) \stackrel{\text{②}}{=} \Phi^{(K)} - \frac{\eta}{W^+}\Phi^{(K)}\sum_{i=1}^{\widetilde{m}} \ell_i^{(K)} \frac{\Lambda_i^{(K)}}{\|\Lambda^{(K)}\|_1}$$

$$= \Phi^{(K)}\left(1 - \frac{\eta}{W^+}\langle \ell^{(K)}, \frac{\Lambda^{(K)}}{\|\Lambda^{(K)}\|_1}\rangle\right) \stackrel{\text{③}}{\leq} \Phi^{(K)} \exp\left(-\frac{\eta}{W^+}\langle \ell^{(K)}, \frac{\Lambda^{(K)}}{\|\Lambda^{(K)}\|_1}\rangle\right)$$

$$\stackrel{\text{④}}{\leq} \Phi^{(1)} \exp\left(-\frac{\eta}{W^+}\sum_{k=1}^{K}\langle \ell^{(k)}, \frac{\Lambda^{(k)}}{\|\Lambda^{(k)}\|_1}\rangle\right) = \widetilde{m} \cdot \exp\left(-\frac{\eta}{W^+}\sum_{k=1}^{K}\langle \ell^{(k)}, \frac{\Lambda^{(k)}}{\|\Lambda^{(k)}\|_1}\rangle\right). \tag{25}$$

Here ① is due to the MW update rule and ② uses $\Phi^{(K)} = \|\Lambda^{(K)}\|_1$. Now ③ uses $1 - x \leq e^{-x}$, for all $x \in \mathbb{R}$. We recursively applied all of the previous inequalities to obtain ④.

On the other hand, we can lower bound

$$\Phi^{(K+1)} \stackrel{\text{①}}{\geq} \Lambda_i^{(K+1)} \stackrel{\text{②}}{=} \Lambda_i^{(1)} \prod_{k=1}^{K}\left(1 - \frac{\eta}{W^+}\ell_i^{(k)}\right) \tag{26}$$

$$\stackrel{\text{③}}{\geq} (1-\eta)^{\frac{1}{W^+}\sum_{\{k:\ell_i^{(k)}\geq 0\}}\ell_i^{(k)}} \cdot (1+\eta)^{\frac{1}{W^+}\sum_{\{k:\ell_i^{(k)}<0\}}\ell_i^{(k)}},$$

where ① holds by the definition of $\Phi^{(K+1)}$ as $\|\Lambda^{(K+1)}\|_1$. Here, ② uses the MW update rule and ③ is due to Bernoulli's inequality: $1 + rx \geq (1 + x)^r$, for $-1 \leq x$, $0 \leq r \leq 1$, with $(x, r) \in \{(-\eta, \ell_i^{(k)}/W^+), (\eta, -\ell_i^{(k)}/W^+)\}$.

Combining (25) and (26), taking logarithms, and multiplying by $\frac{W^+}{\eta}$ we obtain the following inequality ①, which we further bound:

$$\frac{W^+ \log(\widetilde{m})}{\eta} - \sum_{k=1}^{K}\langle \ell^{(k)}, \frac{\Lambda^{(k)}}{\|\Lambda^{(k)}\|_1}\rangle \stackrel{\text{①}}{\geq} \frac{1}{\eta}\log(1-\eta)\left(\sum_{\{k:\ell_i^{(k)}\geq 0\}}\ell_i^{(k)}\right) - \frac{1}{\eta}\log(1+\eta)\left(\sum_{\{k:\ell_i^{(k)}<0\}}\ell_i^{(k)}\right)$$

$$\stackrel{\text{②}}{\geq} (-1-\eta)\left(\sum_{\{k:\ell_i^{(k)}\geq 0\}}\ell_i^{(k)}\right) + (-1+\eta)\left(\sum_{\{k:\ell_i^{(k)}<0\}}\ell_i^{(k)}\right)$$

$$\stackrel{\text{③}}{\geq} -2\eta W^- K + (-1 - \text{sign}(\sigma,\tau)\eta)\sum_{k=1}^{K}\ell_i^{(k)}.$$

In ②, we used $\log(1-\eta) \geq -\eta - \eta^2$ and $\log(1+\eta) \geq \eta - \eta^2$ for $\eta \leq 1/2$. For ③, we have two cases. If $\sigma > \tau$ we have $\text{sign}(\sigma,\tau) = -1$ and we use $-2\eta\ell_i^{(k)} \geq -2\eta\tau = -2\eta W^-$ and bound $-\sum_{k=1}^{K}\mathbb{1}_{\{k:\ell_i^{(k)}\geq 0\}} \geq -K$. Otherwise, we use $2\eta\ell_i^{(k)} \geq -2\eta\sigma = -2\eta W^-$ and bound $-\sum_{k=1}^{K}\mathbb{1}_{\{k:\ell_i^{(k)}<0\}} \geq -K$. Reorganizing terms, dividing by $K$, and using $u = e_i$ we obtain

$$\frac{1}{K}\sum_{k=1}^{K}\langle \ell^{(k)}, \frac{\Lambda^{(k)}}{\|\Lambda^{(k)}\|_1}\rangle \leq \frac{W^+ \log(\widetilde{m})}{\eta K} + 2\eta W^- + \frac{1 + \text{sign}(\sigma,\tau)\eta}{K}\sum_{k=1}^{K}\langle \ell^{(k)}, u\rangle,$$

and finally substituting the value of $\eta = \frac{\delta}{4W^-}$ and $K = \frac{8\sigma\tau \log(\widetilde{m})}{\delta^2} = \frac{8W^+W^- \log(\widetilde{m})}{\delta^2}$ in the statement we obtain the desired result:

$$\frac{1}{K}\sum_{k=1}^{K}\langle \ell^{(k)}, \frac{\Lambda^{(k)}}{\|\Lambda^{(k)}\|_1}\rangle \leq \frac{\delta}{2} + \frac{\delta}{2} + \frac{1 + \text{sign}(\sigma,\tau)\eta}{K}\sum_{k=1}^{K}\langle \ell^{(k)}, u\rangle.$$

$\square$

*Proof* (Lemma 3.3). Since we assumed $\varepsilon < 4\min\{\tau, \sigma\}$, the assumption on $\delta$ required by Lemma C.1 is satisfied: $\delta = \varepsilon/2 < 2\min\{\tau, \sigma\}$. The dimension of the statement was $\widetilde{m} = m$, but we assume nothing on $\widetilde{m}$, so in the proof below it can be that $\widetilde{m} = |I_t|$, i.e., the dimension we obtain when we filter some constraints in Algorithm 2. We would need to substitute the instances of $A$ by $A_{I_t}$. With the parameters of the statement, we obtain the following inequality for any $u \in \Delta^m$:

$$0 \overset{\textcircled{1}}{\leq} \frac{1}{K}\sum_{k=1}^{K}\langle \mathbb{1}_m - Ap^{(k)}, \frac{\Lambda^{(k)}}{\|\Lambda^{(k)}\|_1}\rangle \overset{\textcircled{2}}{\leq} \frac{\varepsilon}{2} + \frac{1+\text{sign}(\sigma,\tau)\eta}{K}\sum_{k=1}^{K}\langle \mathbb{1}_m - Ap^{(k)}, u\rangle, \qquad (27)$$

where $\textcircled{1}$ is satisfied by the oracle assumption (6), after using the fact that $\langle \mathbb{1}_m, \Lambda^{(k)}/\|\Lambda^{(k)}\|_1\rangle = 1$. Inequality $\textcircled{2}$ is the guarantee of the MW algorithm, cf. Lemma C.1. We finally obtain the guarantee we had to prove:

$$-\varepsilon \overset{\textcircled{1}}{\leq} -\frac{\varepsilon}{2(1+\text{sign}(\sigma,\tau)\eta)} \overset{\textcircled{2}}{\leq} 1 - \langle A_i, \bar{p}\rangle, \qquad \text{for all } i \in [m],$$

where $\bar{p} \overset{\text{def}}{=} \frac{1}{K}\sum_{k=1}^{K} p^{(k)}$. Here $\textcircled{1}$ holds regardless of the value of $\text{sign}(\sigma,\tau)$, due to the choice of $\eta \leq 1/2$. Furthermore, $\textcircled{2}$ is obtained by setting $u = e_i$ for all $i \in [m]$ and simplifying (27). $\qquad\square$

*Proof* (Theorem 3.4). Our aim is to use the PST framework to obtain a $\widehat{\lambda} \in \Delta^m$ such that $p^{\widehat{\lambda}}$ is an $(\varepsilon/n)$-minimizer of $\hat{g}$ so that we can use Lemma 3.1 to conclude $\widehat{\lambda}$ is an $\varepsilon$-minimizer of $g$. We will run the MW algorithm several times, restarting the weights between executions. Let $\varepsilon_t \overset{\text{def}}{=} 2^{1-t}$ for $t \geq 0$. At phase $t$, we aim to obtain a dual point $\bar{\lambda}^{(t+1)}$ such that $p^{\bar{\lambda}^{(t+1)}}$ is an $\varepsilon_t$-minimizer of $\hat{g}$. So we first aim for a 2-minimizer and then we halve the accuracy sequentially. The initial point is $\bar{\lambda}^{(0)} \overset{\text{def}}{=} \text{concat}(\mathbb{1}_n/n, \mathbb{0}_{m-n})$, where concat is defined as the concatenation of vectors, so that $\bar{\lambda}^{(0)} = (\frac{1}{n}, \ldots, \frac{1}{n}, 0, \ldots, 0) \in \Delta^m$, with $n$ entries with value $\frac{1}{n}$. This point satisfies that $h^{\bar{\lambda}^{(0)}} = \mathbb{1}_n/n$ and $p^{\bar{\lambda}^{(0)}} = \mathbb{1}_n$, since we assumed $x_i \leq 1$ are the first $n$ constraints of $A$, i.e., the first $n$ rows of $A$ are $e_i$ for $i \in [n]$. Because the maximum entry of $A$ is 1, we have $\hat{g}(p^{\bar{\lambda}^{(0)}}) \leq n$, i.e., $p^{\bar{\lambda}^{(0)}}$ is an $(n-1)$-minimizer of $\hat{g}$. Hence we denote $\varepsilon_{-1} = n-1$ for convenience.

So for $t = 0$, the first phase, we seek to find a 2-minimizer of $\hat{g}$. Thus, this is the only phase in which we use the second case of the value of $\sigma_\delta$, cf. (7). We have $\tau_{\varepsilon_{-1}} = 1$ and $\sigma_{\varepsilon_{-1}} = 2n-2$ and $4\min\{\tau_{\varepsilon_{-1}}, \sigma_{\varepsilon_{-1}}\} = 4$. Note that if $\varepsilon/n \geq 2$, we can actually stop earlier so $T$ in the algorithm is 0, there is only one phase, and $\varepsilon_0$ is actually set to $\varepsilon/n$. At this phase our *good previous solution* is the initial point $\bar{\lambda}^{(0)}$. Assume first $\varepsilon_0 = 2$. The condition $2 = \varepsilon_0 \leq 4\min\{\tau_{\varepsilon_{-1}}, \sigma_{\varepsilon_{-1}}\} = 4$ is satisfied. Thus, according to Lemma 3.3, we reach the $\varepsilon_0$-minimizer after $K_0 = \widetilde{O}(\frac{n}{\varepsilon_0^2}) = \widetilde{O}(n)$ iterations. If $\varepsilon_0$ is $\varepsilon/n > 2$, we can artificially set $\tau_{\varepsilon_{-1}}$ to a larger value, say $\varepsilon/n$, so that the condition $\varepsilon_0 \leq 4\min\{\tau_{\varepsilon_{-1}}, \sigma_{\varepsilon_{-1}}\} = 4\tau_{\varepsilon_{-1}}$ trivially holds, and the complexity is $\widetilde{O}(\frac{\varepsilon/n\cdot\sigma_{\varepsilon_{-1}}}{(\varepsilon/n)^2}) = \widetilde{O}(\frac{n^2}{\varepsilon})$ iterations, which satisfies the theorem. In fact, for large $\varepsilon$, i.e., for $\varepsilon/n > 2$, just aiming for an $\widehat{\varepsilon} = \exp(\varepsilon/n) - 1$ is significantly faster and enough. The latter is true according to Lemma 3.1 without bounding $\log(1 + \widehat{\varepsilon}) \leq \widehat{\varepsilon}$.

Now, if $T > 0$, we run several phases of the MW algorithm. Iteration $t > 0$ takes $K_t \overset{\text{def}}{=} \frac{32\tau_{\varepsilon_{t-1}}\sigma_{\varepsilon_{t-1}}\log(|I_t|)}{\varepsilon_t^2} = \widetilde{O}(\frac{\tau_{\varepsilon_{t-1}}\sigma_{\varepsilon_{t-1}}}{\varepsilon_t^2})$ iterations by the PST guarantee, cf Lemma 3.3. If $\varepsilon_{t-1} > \frac{1}{n}$ we have $\tau_{\varepsilon_{t-1}} \cdot \sigma_{\varepsilon_{t-1}} = O(1 \cdot \varepsilon_{t-1}n)$ and if $\varepsilon_{t-1} \leq \frac{1}{n}$ we have $\tau_{\varepsilon_{t-1}} \cdot \sigma_{\varepsilon_{t-1}} = O(\sqrt{\varepsilon_{t-1}n} \cdot \sqrt{\varepsilon_{t-1}n})$. In any case, it is $K_t = \widetilde{O}(\frac{n}{\varepsilon_t})$, as $\varepsilon_{t-1}/\varepsilon_t = 2 = O(1)$. The assumption $\varepsilon_t \leq 4\min\{\sigma_{\varepsilon_{t-1}}, \tau_{\varepsilon_{t-1}}\}$ is satisfied in these phases. Indeed, if $\varepsilon_t \geq 1/n$, we have $\varepsilon_t < \varepsilon_0 = 2$ and $\sigma_{\varepsilon_{t-1}} \geq 1$, $\tau_{\varepsilon_{t-1}} = 1$. In the case $\varepsilon_t < 1/n$, we have $4\min\{\tau_{\varepsilon_{t-1}}, \sigma_{\varepsilon_{t-1}}\} \geq \sqrt{\varepsilon_t n}$ which is $> \varepsilon_t$.

If $T = \lceil \log_2(2/(\varepsilon/n))\rceil > 0$, then $\varepsilon_T \leq \varepsilon/n$ and the total number of iterations is

$$\sum_{t=0}^{T} K_t = \widetilde{O}(\sum_{t=0}^{T}\frac{n}{\varepsilon_t}) = \widetilde{O}(\sum_{t=0}^{T} n2^{t-1}) = \widetilde{O}(n2^T) = \widetilde{O}(\frac{n^2}{\varepsilon}).$$

The complexity of an iteration is $\widetilde{O}(N)$, assuming $N \geq m$ (recall $m \geq n$ since we added the constraints $x_i \leq 1$). Indeed, computing $I_t$, Algorithm 2, and computing the constraints of the query $\Lambda^{(k)}/\|\Lambda^{(k)}\|_1$ and current solution $\bar{\lambda}^{(t)}$, to be used by the oracle, requires multiplying a vector by $A$ or a subset of its rows, which is $O(N)$. The oracle query takes $\widetilde{O}(n)$, and the rest of operations in Algorithm 2 are simple and take $O(m)$ time. Note the amortized complexity of Algorithm 2 is $O(m)$ per iteration. $\qquad\square$

## C.1 Missing proofs from Section 3.3

The first lemma shows that the oracle returns a point in the lens $\mathcal{L}_{\omega\delta}(v)$ efficiently.

**Lemma C.2.** *Let $v \overset{\text{def}}{=} c(s)/(1+\delta) \in \mathcal{P}$. The feasibility oracle returns a point in the intersection $\mathcal{L}_{\omega\delta}(v) \cap \{x \in \mathbb{R}^n_{\geq 0} : \langle q, x \rangle \leq 1\}$ in time $O(n \log(\frac{n}{(\omega-1)\delta} + \frac{n}{\omega-1}))$.*

*Proof.* The output point $o$ has to satisfy $\textcircled{1} : \langle c^{-1}(o), v \rangle \leq 1$ and $\textcircled{2} : \langle c^{-1}(v), o \rangle \leq 1 + \omega\delta$ to be in the lens and $\textcircled{3} : \langle q, o \rangle \leq 1$ to be in the halfspace defined by $q$.

We note that the definition of $c(\cdot)$ implies $c^{-1}(v) = (1+\delta)s$. Condition $\textcircled{1}$ is always trivially satisfied because $c^{-1}(o) \in \mathcal{D}$ and $v \in \mathcal{P}$ by construction.

Now we have three cases. If $\langle s, c(q) \rangle \leq \frac{1+\omega\delta}{1+\delta}$ the oracle returns $o = c(q)$ and $\lambda^{(o)} = \lambda^{(q)}$. This satisfies the other two conditions. Indeed, $\textcircled{3}$ comes from $\langle q, c(q) \rangle = 1$ and $\textcircled{2}$ is satisfied because $\langle s, c(q) \rangle \leq \frac{1+\omega\delta}{1+\delta}$ implies $\langle c^{-1}(v), c(q) \rangle = \langle (1+\delta)s, c(q) \rangle \leq 1 + \omega\delta$. If we have $\langle q, c(s) \rangle \leq 1$, then the oracle returns $o = c(s)$ and $\lambda^{(o)} = \lambda^{(s)}$. In this case $\textcircled{3}$ is satisfied by construction, and $\textcircled{2}$ is satisfied because $\langle c^{-1}(v), c(s) \rangle = \langle (1+\delta)s, c(s) \rangle = 1 + \delta \leq 1 + \omega\delta$.

From now on we may focus on the third case, where $\langle s, c(q) \rangle > \frac{1+\omega\delta}{1+\delta}$ and $\langle q, c(s) \rangle > 1$. Let us define the functions $\pi_s, \pi_q : (0,1) \to \mathbb{R}_{\geq 0}$ as:

$$\begin{aligned} \pi_s(\mu) &= \langle s, c((1-\mu)s + \mu q) \rangle, \\ \pi_q(\mu) &= \langle q, c((1-\mu)s + \mu q) \rangle. \end{aligned}$$

The key observation relating these two functions is $(1-\mu)\pi_s(\mu) + \mu\pi_q(\mu) = 1$ for any $\mu \in (0,1)$ because $\langle h, c(h) \rangle = 1$ for any constraint $h \in \mathbb{R}_{\geq 0}$. So, if we find a $\mu^* \in (0,1)$ with $\pi_s(\mu^*) \in (1, \frac{1+\omega\delta}{1+\delta})$ then $o = c((1-\mu^*)s + \mu^* q)$ will satisfy both $\textcircled{2}$, because of $\pi_s(\mu^*) < (1+\omega\delta)/(1+\delta)$, and also $\textcircled{3}$, because if $\pi_s(\mu^*) > 1$ then $\pi_q(\mu^*) < 1$ by the observation. And we recover $\lambda^{(o)}$ as $(1-\mu^*)\lambda^{(s)} + \mu^* \lambda^{(q)}$.

We intend to find such a $\mu^*$ with the bisection method. Despite $\pi_s$ having a potential singularity near $\mu = 1$, we will show it is regular enough to guarantee fast convergence. By the assumptions, $\lim_{\mu\to 1} \pi_s(\mu) > \frac{1+\omega\delta}{1+\delta}$ and $\lim_{\mu\to 0} \pi_q(\mu) > 1$. Then, $\pi_q(\mu) > 1$ for any $\mu$ small enough, which means $\pi_s(\mu) < 1$ for any $\mu$ small enough by the observation. Finally, $\pi_s$ is continuous, so we are able to find $\mu^*$ with $\pi_s(\mu^*) \in (1, \frac{1+\omega\delta}{1+\delta})$ via the bisection method. The only remaining question is computing its running time, for which we lower bound the length of an interval in $(0,1)$ that satisfies the conditions.

For that we will bound $\pi'_s(\mu)$. Let us start with the definition of $\pi'_s(\mu)/\pi_s(\mu)$. Let $\pi_s(\mu)_i \overset{\text{def}}{=} \frac{s_i}{n((1-\mu)s_i + \mu q_i)}$ be the $i$-th summand in the inner product of $\pi_s(\mu)$. We have

$$\frac{\pi'_s(\mu)}{\pi_s(\mu)} = \frac{\sum_{i\in[n]} \frac{s_i n(s_i - q_i)}{n^2((1-\mu)s_i + \mu q_i)^2}}{\sum_{i\in[n]} \pi_s(\mu)_i} = \frac{\sum_{i\in[n]} \pi_s(\mu)_i \frac{(s_i - q_i)}{(1-\mu)s_i + \mu q_i}}{\sum_{i\in[n]} \pi_s(\mu)_i}.$$

We have $\pi_s(\mu)_i \geq 0$ so the expression above is a weighted arithmetic mean, and its value is at most that of the maximum of the summands:

$$\frac{\pi'_s(\mu)}{\pi_s(\mu)} \leq \max_{i\in[n]} \frac{s_i - q_i}{(1-\mu)s_i + \mu q_i} \overset{\textcircled{1}}{\leq} n \max_{i\in[n]} \pi_s(\mu)_i \leq n \sum_{i\in[n]} \pi_s(\mu)_i \leq n\pi_s(\mu).$$

We dropped $-q_i/((1-\mu)s_i + \mu q_i)$ in ① above. Hence, we have $\pi_s'(\mu) \leq n\pi_s^2(\mu)$. This means the preimage of $J \stackrel{\text{def}}{=} (1, \frac{1+\omega\delta}{1+\delta})$ is an interval of length at least

$$\frac{\frac{1+\omega\delta}{1+\delta} - 1}{\max_{\mu:\pi_s(\mu)\in J} \pi_s'(\mu)} \geq \frac{\frac{1+\omega\delta}{1+\delta} - 1}{\max_{\mu:\pi_s(\mu)\in J} n\pi_s^2(\mu)} = \frac{\frac{1+\omega\delta}{1+\delta} - 1}{n\left(\frac{1+\omega\delta}{1+\delta}\right)^2}.$$

We are interested in upper bounding the inverse of the length:

$$\frac{n\left(\frac{1+\omega\delta}{1+\delta}\right)^2}{\frac{1+\omega\delta}{1+\delta} - 1} = \frac{n(1+\omega\delta)^2}{(\omega-1)\delta(1+\delta)} = n\left(\frac{1}{(\omega-1)\delta} + \frac{2\omega-1}{\omega-1} + \frac{\delta(\omega-1)}{1+\delta}\right) \leq \frac{n}{(\omega-1)\delta} + \frac{4n}{\omega-1}.$$

Since the bisection method starts with an interval of length 1 and progressively halves it every iteration, it takes at most $\log_2\left(\frac{n}{(\omega-1)\delta} + \frac{4n}{\omega-1}\right)$ iterations to find a point with $\pi_s(\mu^*) \in (1, \frac{1+\omega\delta}{1+\delta})$, and each step takes $O(n)$ in processing time. Thus, the oracle returns a point in time $O(n\log(\frac{n}{(\omega-1)\delta} + \frac{n}{\omega-1}))$. $\square$

The following lemma bounds the lens by a box defined in terms of $\omega\delta$.

**Lemma C.3.** *If $x \in \mathcal{L}_{\omega\delta}(v)$, with $v \in \mathbb{R}^n_{\geq 0}$, $\omega\delta > 0$, then,*

$$\begin{aligned}
x_i &\geq L_i \stackrel{\text{def}}{=} \max(0, 1 - \sqrt{\omega\delta n})v_i, \\
x_i &\leq U_i \stackrel{\text{def}}{=} (1 + \sqrt{\omega\delta n} + \omega\delta n)v_i.
\end{aligned}$$

*We call the region $\prod_i [L_i, U_i]$ the bounding box of the lens.*

*Proof* (Lemma C.3). Recall that the lens is defined as the set of points $x \in \mathbb{R}^n_{\geq 0}$ satisfying both $\langle c^{-1}(v), x \rangle \leq 1 + \omega\delta$ and $\langle c^{-1}(x), v \rangle \leq 1$. Let us rewrite these conditions as sums:

$$\begin{cases}
\langle c^{-1}(v), x \rangle = \sum_{i\in[n]} \frac{x_i}{nv_i} &\leq 1 + \omega\delta, \\
\langle c^{-1}(x), v \rangle = \sum_{i\in[n]} \frac{v_i}{nx_i} &\leq 1.
\end{cases}$$

These two constraints are invariant up to multiplications of $x_i$ and $v_i$ by the same constant. Let $z_i = x_i/v_i$, and multiply both by $n$ to get:

$$\begin{cases}
\sum_{i\in[n]} z_i &\leq (1 + \omega\delta)n, \\
\sum_{i\in[n]} z_i^{-1} &\leq n.
\end{cases}$$

It is our purpose to find the maximum and minimum of $z_i$ in this region. Because the system is symmetric under reordering of the $z_i$, we may focus on bounding $z_1$. Since the region is convex and symmetric under reordering of the variables, and because the function $z \mapsto z_1$ is symmetric under reordering of the last $(n-1)$ variables, we may also assume that the maximum and minimum of this function are attained in points with $z_2 = \cdots = z_n$.

This brings us to:

$$\begin{cases}
z_1 + (n-1)z_2 &\leq (1+\omega\delta)n, \\
z_1^{-1} + (n-1)z_2^{-1} &\leq n.
\end{cases}$$

The two constraints independently will never have normals vectors proportional to $e_1$. Furthermore the feasible region of the second constraint in $\mathbb{R}^n_{\geq 0}$ is contained in the interior of $\mathbb{R}^n_{\geq 0}$. This means that the solutions maximizing and minimizing $z_1$ must satisfy both constraints with equality.

Solving the system of equations gives two roots. The two solutions for $z_1$ are:

$$z_1^+ = \frac{1}{2}(\omega\delta n + 2 + \sqrt{\omega^2\delta^2 n^2 + 4\omega\delta n - 4\omega\delta}),$$

$$z_1^- = \frac{1}{2}(\omega\delta n + 2 - \sqrt{\omega^2\delta^2 n^2 + 4\omega\delta n - 4\omega\delta}).$$

Let us bound the smaller one first. We give two such lower bounds. The trivial one is that $z_1^- > 0$. This comes from the fact that the second constraint already guarantees $z_i > 0$.

The second lower bound comes from $\sqrt{a+b} \leq \sqrt{a} + \sqrt{b}$ for $a, b > 0$, a consequence of the triangle inequality:

$$z_1^- = \frac{1}{2}(\omega\delta n + 2 - \sqrt{\omega^2\delta^2 n^2 + 4\omega\delta n - 4\omega\delta}) \geq \frac{1}{2}(\omega\delta n + 2 - \sqrt{\omega^2\delta^2 n^2 + 4\omega\delta n}) \geq$$
$$\geq \frac{1}{2}(\omega\delta n + 2 - \omega\delta n - 2\sqrt{\omega\delta n}) \geq 1 - \sqrt{\omega\delta n}.$$

Now let us study the larger root. As with the other root, we remove the $-4\omega\delta$ term in the square root, then apply the triangle inequality:

$$z_1^+ = \frac{1}{2}(\omega\delta n + 2 + \sqrt{\omega^2\delta^2 n^2 + 4\omega\delta n - 4\omega\delta}) \leq \frac{1}{2}(\omega\delta n + 2 - \sqrt{\omega^2\delta^2 n^2 + 4\omega\delta n}) \leq$$
$$\leq \frac{1}{2}(\omega\delta n + 2 + \omega\delta n + 2\sqrt{\omega\delta n}) \geq 1 + \omega\delta n + \sqrt{\omega\delta n}.$$

Undoing the change of variables $z_i = x_i/v_i$ we obtain the desired bounds. $\qquad\square$

Observe that indeed the optimum $p^{\lambda^*}$ satisfies the two conditions in the definition of $\mathcal{L}_{\omega\delta}(v)$: The first condition $\langle c^{-1}(p^{\lambda^*}), v \rangle \leq 1$ comes from $p^{\lambda^*} \in c(\mathcal{D})$, so $c^{-1}(p^{\lambda^*}) \in \mathcal{D}$ covers $\mathcal{P}$, i.e., $\langle c^{-1}(p^{\lambda^*}), x \rangle \leq 1$ for all $x \in \mathcal{P}$. In particular it covers $v$. The second condition $\langle c^{-1}(v), p^{\lambda^*} \rangle \leq 1 + \omega\delta$ is equivalent to $\langle \frac{1+\delta}{1+\omega\delta} s, p^{\lambda^*} \rangle \leq 1$. Since $\omega > 1$, this is satisfied as $p^{\lambda^*} \in \mathcal{P}$ and $s \in \mathcal{D}$ and therefore $\frac{1+\delta}{1+\omega\delta} s \in \mathcal{D}^+$. The last two lemmas provide the intuition of why the oracle returns points that are not too far from $p^{\lambda^*}$: for a fixed $\omega > 1$ the bounding boxes of the respective lenses get smaller as $\delta \to 0$.

Now we can finally prove the exact guarantees of the oracle. Let $q = A^T \lambda^{(q)}$ with $\lambda^{(q)} \in \Delta^m$ be the query. Furthermore, let $s = A^T \lambda^{(s)}$ with $\lambda^{(s)} \in \Delta^m$ be our current good solution, so that the point $c(s) = c(A^T \lambda^{(s)})$ satisfies $\hat{g}(c(s)) \leq 1 + \delta$. The following proposition proves the guarantees on the oracle.

**Proposition C.4.** *Let $U \overset{\text{def}}{=} c(s)(1 + 2\delta n + \sqrt{2\delta n})/(1+\delta)$ be the upper-most vertex of the bounding box of the lens $\mathcal{L}_{\omega\delta}(v)$, as defined in [Lemma C.3]. Let $I$ be the set of non-redundant constraints, defined as $I \overset{\text{def}}{=} \{i \in [m] : \langle A_i, U \rangle \geq 1\}$. Furthermore, let the following be the width parameters of the oracle $\mathfrak{O}$ implemented in [Algorithm 3]:*

$$\sigma \overset{\text{def}}{=} \min(\sqrt{\omega\delta n} + \omega\delta n, \frac{1+\omega\delta}{1+\delta}\max_{i\in[n]}\frac{1}{s_i} - 1) \qquad and \qquad \tau \overset{\text{def}}{=} \min(3\sqrt{\omega\delta n}, 1). \qquad (28)$$

*Then, the oracle $\mathfrak{O}$ in [Algorithm 3] returns a pair $\lambda^{(o)} \in \Delta^m$, $o \in c(\mathcal{D})$ such that*

1. *$o$ satisfies $q$, i.e., $\langle q, o \rangle \leq 1$.*

2. *If $i \in I$, it yields $\langle A_i, o \rangle \in [1-\tau, 1+\sigma]$. That is, it is compatible with the width parameters $\sigma, \tau$ as above, with the loss $1 - \langle A_i, o \rangle$ in $[-\sigma, \tau]$.*

3. *$o$ satisfies all redundant constraints, i.e., $\langle A_i, o \rangle \leq 1$, if $i \in [m] \setminus I$.*

*Besides, $o = c(A^T \lambda^{(o)})$, and [Algorithm 3] runs in time $O(n \log(\frac{n}{(\omega-1)\delta} + \frac{n}{\omega-1}))$.*

*Proof.* The first claim is proven in [Lemma C.2].

Let us consider $I$ now. It is clear that the constraints in $I$ are exactly those that do not cover the bounding box of [Lemma C.3] around the lens $\mathcal{L}_{\omega\delta}(v)$, with $v \overset{\text{def}}{=} c(s)/(1+\delta)$. This is because a positive constraint covers a box if and only if it covers the upper-most vertex $U$. Since the oracle returns a point in the lens, and hence in the box, any constraint not in $I$ is automatically satisfied by any point returned by the oracle. This is the third claim.

Note that the geometric meaning of the second claim is that $o$ is close to be lying on the hyperplanes $\langle A_i, x \rangle = 1$. If $i \in I$, then $\langle A_i, U \rangle \geq 1$ by the definition of $I$. Define $L$ as the lower-most vertex of the bounding box of $\mathcal{L}_{\omega\delta}(v)$. Now,

$$\langle A_i, o \rangle \geq \min_{x\in\mathcal{L}_{\omega\delta}(v)} \langle A_i, x \rangle \geq \min\{\langle A_i, x \rangle : x \in \mathbb{R}^n_{\geq 0}, x_i \in [L_i, U_i]\}.$$

And the second minimum will be attained in the lower-most vertex as $A_{ij} \geq 0$ for all $j \in [n]$. Therefore:

$$\langle A_i, o \rangle \geq \min_{x \in L(v, \omega\delta)} \langle A_i, x \rangle \geq \langle A_i, L \rangle = \langle A_i, U \rangle \frac{\max(0, 1 - \sqrt{\omega\delta n})}{1 + \sqrt{\omega\delta n} + \omega\delta n}$$

$$\geq \frac{\max(0, 1 - \sqrt{\omega\delta n})}{1 + \sqrt{\omega\delta n} + \omega\delta n} = 1 - \min\left(1, \frac{2\sqrt{\omega\delta n} + \omega\delta n}{1 + \sqrt{\omega\delta n} + \omega\delta n}\right).$$

The second argument of the $\min$ is only less than $1$ whenever $\omega\delta n < 1$, so we may assume the latter in order to obtain the following bound

$$\langle A_i, o \rangle \geq \min_{x \in L(v, \omega\delta)} \langle A_i, x \rangle \geq 1 - \min\left(1, \frac{2\sqrt{\omega\delta n} + \omega\delta n}{1 + \sqrt{\omega\delta n} + \omega\delta n}\right) \geq 1 - \min(1, 3\sqrt{\omega\delta n}) = 1 - \tau.$$

Finally, we focus on the bound with $\sigma$. As before, the maximum of $\langle A_i, x \rangle$ will be attained at a vertex of the box, only this time it is $U$. Now, as $v \in \mathcal{P}$, we have $\langle A_i, v \rangle \leq 1$. We use these two facts to conclude:

$$\langle A_i, o \rangle \leq \max_{x \in L(v, \omega\delta)} \langle A_i, x \rangle \leq \langle A_i, U \rangle = \langle A_i, v \rangle (1 + \sqrt{\omega\delta n} + \omega\delta n) \leq 1 + \sqrt{\omega\delta n} + \omega\delta n.$$

The second upper bound on $\langle A_i, o \rangle$ does not come from the bounding box. We only look at the linear component of the lens, and since $A_i$ is positive, it must attain a maximum in one of the vertices of the simplex in the intersection of the hyperplane with the positive orthant. We also use that $A_{ij} \leq 1$:

$$\langle A_i, o \rangle \leq \max_{x \in \mathcal{L}_{\omega\delta}(v)} \langle A_i, x \rangle \leq \max_{\substack{x \in \mathbb{R}^n_{\geq 0} \\ \langle s, x \rangle \leq (1+\omega\delta)/(1+\delta)}} \langle A_i, x \rangle = \max_{j \in [n]} A_{ij} \frac{1 + \omega\delta}{1 + \delta} \frac{1}{s_j} \leq \frac{1 + \omega\delta}{1 + \delta} \max_{j \in [n]} \frac{1}{s_j}.$$

Thus, $\langle A_i, o \rangle \leq 1 + \min(\sqrt{\omega\delta n} + \omega\delta n, \frac{1+\omega\delta}{1+\delta} \max_{j \in [n]} \frac{1}{s_j} - 1) = 1 + \sigma$.

The running time and the fact $o = c(A^T \lambda^{(o)})$ follow from Lemma C.2. $\qquad\square$