# OpenReview forum: "Fast Algorithms for Packing Proportional Fairness and its Dual"
_NeurIPS.cc/2022/Conference — NeurIPS 2022 Accept_

### Official Review · Reviewer_nmUR · 2022-07-07

**Rating:** 5
**Confidence:** 4
**Soundness:** 3 good
**Presentation:** 3 good
**Contribution:** 2 fair

**Summary:**

This paper studies proportional fair resource allocation problem and develops first-order methods

**Questions:**

Can the algorithms be generalized to other optimization/learning problems?

Theorem 2.5 depends on big constants, which can be exponential in n?

**Ethics Review Area:**

["I don’t know"]

**Limitations:**

The developed first-order method is limited to the problem itself.

**Strengths And Weaknesses:**

Strengths: The analysis is solid and the problem is interesting.

Weakness: The developed first-order method is limited to the problem itself.

---

> ### Author Response · Authors · 2022-07-31
> **rebuttal**
>
>
> > Can the algorithms be generalized to other optimization/learning problems?
>
> The algorithms are designed for the two problems we studied and exploit their structure in order to obtain fast convergence that cannot be achieved with general purpose convex optimization algorithms since the smoothness and Lipstchitz constants depend on the width of the problem. We show the dual problem has applications to the very general problem of linear programming and show improvement. Achieving fast and width-independent algorithms for these two problems is of interest, as Reviewer wjuT pointed out.
>
>
> > Theorem 2.5 depends on big constants, which can be exponential in n?
>
> This is not true, the number of iterations is $T = \widetilde{O}(n/\epsilon)$ and each iteration has cost equal to the number of non-zero entries of the matrix A. And the constants are small. Note that in line 3 of Algorithm 1 we write a bound on the value of T with its constants: $T \leq 3L\log(4n\log(2mn)/\epsilon)+1$ where $L$ is written out in line 1.
>
> The $\exp(y^{(T)})$ indicates coordinatewise exponential of the final point $y^{(T)}$ which is used for going from the parametrization of $f_r$ to the one of $f$ and has nothing to do with the complexity of the algorithm.

---

### Official Review · Reviewer_2so1 · 2022-07-11

**Rating:** 5
**Confidence:** 2
**Soundness:** 3 good
**Presentation:** 3 good
**Contribution:** 2 fair

**Summary:**

The paper solves the so-called 1-fair packing problem via first-order methods. The main results are computational complexity of O(n/epsilon) for the primal problem and O(n^2/epsilon) for the dual problem. These results are not width dependent (where the width is the ratio of largest to smallest elements in matrix A), which is a plus.

**Questions:**

I am not too familar with the literature on this problem; perhaps a in-depth discussion about pros and cons of the developed algorithms compared to the state-of-the-art would be helpful to readers.

**Limitations:**

The authors did not address the limitations and potential negative societal impact of their work but it is okay given the nature of this work.

**Strengths And Weaknesses:**

The strength is that the paper studies a fairly important problem in resource allocation. I like the results overall. The weakness is that the method developed follows closely to Allen-Zhu and Orecchia (Math. Prog. 2019). It seems that it simply customized their approach to this specific setting. The computational complexity results are good but at the expense of solving more difficult oracles.

---

> ### Author Response · Authors · 2022-07-31
> **rebuttal**
>
> > The weakness is that the method developed follows closely to Allen-Zhu and Orecchia (Math. Prog. 2019)
>
> Firstly, please note that our method for the primal problem differs in some important aspects from that of Allen-Zhu and Orecchia for packing LPs. As a matter of fact our method is deterministic and distributed (and thus it can be parallelizable as well), whereas in Packing LPs obtaining an accelerated width-independent method that is deterministic or parallelizable is an open question. Also, we have to deal with additive errors, whereas Allen-Zhu and Orecchia aim for multiplicative errors.
>
> Secondly, our work extends far beyond the primal solution: we solve the dual problem with a completely different algorithm and we show a new application of this problem to linear programming.
>
> > The computational complexity results are good but at the expense of solving more difficult oracles.
>
> We do not use more difficult oracles for the primal problem, at each iteration we need to compute the gradient of $f_r$, which only requires a matrix multiplication and some simple operations. The rest of an iteration consists of simple operations as well.
>
> As for the dual, the "difficulty" of the geometric oracle is only in the analysis, the complexity of an iteration of the algorithm for the dual problem is similar to that of the primal. In sum, we require $\widetilde{O}(n^2/\epsilon)$ (parallelizable and possibly sparse) matrix multiplications and that same order of simple operations.
>
>
> > Perhaps an in-depth discussion about pros and cons of the developed algorithms compared to the state-of-the-art would be helpful to readers.
>
> Thanks for the suggestion, we will make sure to emphasize the comparisons. Here is a summary: For the primal, we discussed how the state of the art was [DFO20] with complexity $\widetilde{O}(n^2/\epsilon^2)$ and polylog dependence on the width. Our algorithm is strictly better as it works under the same assumptions and has a complexity of $\widetilde{O}(n/\epsilon)$ with no dependence on the width. For the dual problem, we obtain the first width-independent algorithm. For the application to the Yamnitsky-Levin algorithm we provided a discussion and analysis showing improvement over this algorithm.
>
> Please, do not hesitate to ask any other questions you may have about our work, we will be happy to answer them. Thank you for your time reviewing our work.

---

### Official Review · Reviewer_wjuT · 2022-07-11

**Rating:** 7
**Confidence:** 4
**Soundness:** 3 good
**Presentation:** 3 good
**Contribution:** 4 excellent

**Summary:**

**The problem and its importance**

The paper presents an *accelerated, width-independent algorithm* for the $\alpha$-fair-packing problem with $\alpha = 1$, in the distributed model of computation. The problem is motivated by that of fair allocation of (non-negative) resources and falls under the broad umbrella of $\alpha$-fair-packing problems, which are an important problem class encompassing, for example, the standard packing LP, which has seen a long line of work (for example, by Luby-Nisan, Fleischer, Plotkin-Shmoys-Tardos, Awerbuch-Khandekar, Young, Koufogiannakis-Young, Allen-Zhu-Orecchia, Wang, Mahoney-Rao-Wang-Zhang, etc.) in both theoretical computer science (most significantly in the context of multicommodity flow problems) and machine learning that have led to developments of techniques that are now widely used in optimization in general. Therefore, this paper definitely studies a very important problem.

The paper further studies the dual of this algorithm and applies it to obtain an improved analysis of the classical Yamnitsky-Levin algorithm.

**The technical outline**

The paper solves the primal problem by building upon the work of Diakonikolas-Fazel-Orecchia and AllenZhu-Orecchia. Specifically, the problem reformulation and parts of analysis (the gradient descent progress) is drawn from DFO, whereas the framework of ``truncated gradients in mirror descent coupled with gradient descent'' comes from AZO. We explain these a bit more next.

The notion of ``fairness'' is captured in the objective via maximizing the *product* of allocated resources (and taking the log to formulate it as a convex minimization problem), which results in the problem $\max_{x\geq 0, Ax \leq 1} \, \sum_{i = 1}^n \log x_i$. The final reformulated problem involves the following modifications to this problem: (1) reparametrizing the problem variable to make the (initial) objective linear; (2) moving the packing constraints into the objective via an exponential penalty; (3) adding a box constraint based on properties of the optimal point. We note that these modifications were first done in DFO.

For this reformulated problem, the gradients lie (coordinatewise) in $[-1, \infty)$. The algorithm proceeds as follows. It takes a mirror descent step using the squared-loss mirror map and on the linear term defined by the *truncated* gradient. The truncation is essential for obtaining a regret that is independent of the width of the problem, and this idea stems from AZO. Next, the algorithm takes a ``gradient descent step'' with the amount of update scaled proportional to the amount of movement via the mirror descent step. Following the gradient descent analysis of DFO, the paper shows a minimum bound on the progress via this step. Finally, the iterates obtained via the mirror descent and gradient descent steps are coupled in a convex combination (and their analysis performed analogous to that in AZO). This gives a primal algorithm with $\epsilon$-additive optimal point in $O(n/\epsilon)$ iterations.

**Contributions**

1. The paper's main result is an $O(n/\epsilon)$ iteration algorithm for the stated primal problem. The previous best result by DFO was $O(n^2/\epsilon^2)$, making this paper's contribution in terms of results very clearcut. Further, in achieving this iteration count, the paper becomes the first to achieve both width-independence and acceleration for the $1$-fair-packing problem, which is quite a significant and a highly non-trivial contribution in this line of work.

2. While the paper definitely builds upon the work of DFO and AZO (as described above), the way it combines them is novel (which is what gives it its new, improved result).






**Questions:**

0. Please see the **Weaknesses** section above for other questions.

1. My first obvious question is, why doesn't this algorithm extend to the case $\alpha \neq 1$? Or, if it does, then what is the iteration complexity? It would make the paper more complete to explain this as well.

**Limitations:**

I very much appreciate the authors being upfront about the paper not generalizing to the $\alpha \neq 1$ case. Given that this is a theoretical paper, I do not foresee any negative societal impact of the paper and have no suggestions to this end.

**Strengths And Weaknesses:**

**Strengths**

1. Achieving both width-independence and acceleration for $1$-fair-packing is quite a significant result. In my opinion, just the first component of the paper (the primal result) is a strong result in itself.

2. The paper's primal algorithm is very clean, with very clearcut operations (mirror descent, gradient descent, and coupling) along with their corresponding analyses. The *reasons* for various steps in the algorithm (example, gradient truncation) and why the analysis works (example, the progress by gradient descent compensating the regret bound via the truncated gradient) is also all explained very transparently.

3. The previous best result on this problem (DFO) achieves their result by a different technique (approximate dual gap technique), so it is quite nice that this follow-up work follows a different technique.

**Weaknesses**

1. Given that the paper greatly builds upon the work of DFO (for the primal algorithm), I think one way to make this paper even more transparent would be to (if possible) try and frame DFO's algorithm in this paper's language. More concretely, can you write DFO's algorithm in the ``coupling'' framework of this paper (which in turn draws this from AZO) and show exactly what steps of DFO (in this framework) were changed to achieve this paper's result?

2. I think the main body (the primal section) could do a slightly better job of giving more complete justifications/attributions. As examples,

    * it's not immediate from the algorithm in the main body that the iterates all lie in the box (though this is justified in the appendix); perhaps a pointer to this lemma would be good.

    * Further, I think it's important to clearly state in the main body what parts of the problem formulation and analysis are derived from DFO and AZO (for example, Lemma 2.3 is basically that of DFO). Having these attributions clearly stated will more clearly shine a light on the paper's novelty, without taking away the contributions of DFO and AZO.

3. I found the dual algorithm/analysis way more dense than the primal part. Is it possible to simplify this section's exposition?

**My overall verdict**

I think this is a worthy problem, with a simple, clean algorithm for the primal case, a creative one for the dual case, and an interesting application to a classical algorithm (Yamnitsky-Levin) that I had been unaware of.

In addition to its primary contributions (the primal and dual algorithms and analysis), the final contribution adds value to the optimization/ML literature by bringing to fore a (possibly) less-known classical algorithm. This would therefore be a nice contribution to NeurIPS '22.

---

> ### Author Response · Authors · 2022-07-31
> **rebuttal**
>
>
> Thank you for your time and thorough review. We reply to your questions and comments below and will be happy to answer any further questions.
>
> > Given that the paper greatly builds upon the work of DFO (for the primal algorithm), I think one way to make this paper even more transparent would be to (if possible) try and frame DFO's algorithm in this paper's language. More concretely, can you write DFO's algorithm in the ``coupling'' framework of this paper (which in turn draws this from AZO) and show exactly what steps of DFO (in this framework) were changed to achieve this paper's result?
>
> We do not know how we could do such reformulation, the analysis and algorithm of DFO are quite different.
>
> > it's not immediate from the algorithm in the main body that the iterates all lie in the box (though this is justified in the appendix); perhaps a pointer to this lemma would be good.
>
> We note that we referenced to this in line 136 when introducing the variables x, y, z in section 2.
>
> > I think it's important to clearly state in the main body what parts of the problem formulation and analysis are derived from DFO and AZO (for example, Lemma 2.3 is basically that of DFO). Having these attributions clearly stated will more clearly shine a light on the paper's novelty, without taking away the contributions of DFO and AZO.
>
> We will go through the paper and will make some changes to emphasize in the main paper the part of the analysis that uses a previous result.
>
> > I found the dual algorithm/analysis way more dense than the primal part. Is it possible to simplify this section's exposition?
>
> The algorithm for the dual problem and its ideas are very different from the ones of the primal. We simplified the analysis greatly from our first analysis for this algorithm but we are not aware of anything that could simplify it. The dual problem seems harder to tackle and indeed that is the case as well for packing LP. We had to design a completely different solution using different elements and techniques.
>
> > My first obvious question is, why doesn't this algorithm extend to the $\alpha\neq 1$ case? Or, if it does, then what is the iteration complexity? It would make the paper more complete to explain this as well.
>
> Of the two algorithms, the dual heavily uses the geometry of the problem in order to craft the solution. The primal one also uses some particular structures of this objective in order to find a fast solution and it is not clear how to do so beyond the $\alpha=1$ case.
>
> It should not come as a surprise that the structural properties of the case $\alpha=1$ are different from the others. DFO have unaccelerated algorithms for the primal problem for $\alpha\in[0,1]$ but they needed to provide two algorithms and analyses: one for $\alpha\in[0,1)$ and another one for $\alpha=1$. In sum, the problems' structures are different and finding fast algorithms for this other case is an interesting future direction of research.

---

> > ### Comment · Reviewer_wjuT · 2022-08-08
> > **Thank you for your response!**
> >
> > I am grateful to the authors for clarifying all their points. I will be happy to raise my score to 8 if the authors can show me evidence of their work having "...excellent impact on at least one area of AI or high-to-excellent impact on multiple areas of AI..." that I might have missed (this seems to be the only point of difference between a 7 and an 8).
> >
> > In any case, I am certain that this is a very nice contribution to NeurIPS, and during the discussion period, I will be arguing for it to be accepted.

---

### Official Review · Reviewer_539K · 2022-07-11

**Rating:** 5
**Confidence:** 3
**Soundness:** 3 good
**Presentation:** 3 good
**Contribution:** 3 good

**Summary:**

This paper studies the proportionally fair optimization problem (i.e., $\max f(x) = \sum_i \log x_i$) under packing constraints ($A x \leq 1$ for a matrix A of nonnegative entries). The main contribution is a distributed accelerated algorithm for fair packing. The algorithm is width independent and produces (additively) approximate solutions for both primal and dual.

Width is defined as max ratio of positive entries in the matrix A. First order width-dependent methods don't readily lead to polynomial-time algorithms because smoothness constants do not scale logarithmically with width. In addition, approximate solutions of primal and dual are not easily translatable. In light of this, designing width-independent approximation algorithms for both primal and dual sounds challenging.

This is done by extending a general technique for packing LPs and exploiting problem structure to design a distributed algorithm. Distributed width-independent algorithms for general packing LPs are not yet known. Unlike prior work, the new algorithm is determinstic, has additive (not multiplicative) error, and uses a different regularization. The new algorithm uses $O(n/\epsilon)$ iterations, a factor n smaller than the best prior method.

**Questions:**

none

**Strengths And Weaknesses:**

Strengths:
- Primal and dual algorithms are both width independent. (Translating approximate solutions between primal & dual is not easy.)

Weaknesses:
- No discussion of concrete applications, hence no experimental study to show practical benefits, runtime and error bounds.

---

> ### Author Response · Authors · 2022-07-31
> **rebuttal**
>
> > In light of this, designing width-independent approximation algorithms for both primal and dual sounds challenging. This is done by extending a general technique for packing LPs and exploiting problem structure to design a distributed algorithm.
>
> We note this only applies to the algorithm for the primal problem. The dual problem is tackled with a very different technique.
>
> > No discussion of concrete applications, hence no experimental study to show practical benefits, runtime and error bounds.
>
> We note that we discussed an application to a linear programming algorithm for the dual problem and we provided an analysis showing we can improve the Yamnitsky-Levin algorithm for LP. For the primal problem, we cited several applications. The focus of this work was on designing the algorithms and providing a theoretical analysis.
>
> Do not hesitate to ask any other questions that may arise after this phase, we will be happy to answer them. Thank you very much for your time and effort reviewing our paper.

---

### Meta-Review · Area_Chair_xTJT · 2022-08-24

**Recommendation:** Accept
**Confidence:** Less certain

**Metareview:**

The paper proposes fast algorithms for computing proportional fair allocations, which is one of the most widely defs of fairness. Although reviews were mixed, we believe that importance of the problem makes this a worthy paper to include in NeurIPS. However, we encourage the authors to incorporate the comments of the reviewers to make it more interesting for the ML audience.

**Award:**

No

---

### Decision · Program_Chairs · 2022-09-14

Accept